# Implicit Bias of Gradient Descent for Logistic Regression at the Edge of Stability

**Jingfeng Wu**
Johns Hopkins University
Baltimore, MD 21218
uuujf@jhu.edu

**Vladimir Braverman**
Rice University
Houston, TX 77005
vb21@rice.edu

**Jason D. Lee**
Princeton University
Princeton, NJ 08544
jasonlee@princeton.edu

## Abstract

Recent research has observed that in machine learning optimization, gradient descent (GD) often operates at the *edge of stability* (EoS) [Cohen et al., 2021], where the stepsizes are set to be large, resulting in non-monotonic losses induced by the GD iterates. This paper studies the convergence and implicit bias of constant-stepsize GD for logistic regression on linearly separable data in the EoS regime. Despite the presence of local oscillations, we prove that the logistic loss can be minimized by GD with *any* constant stepsize over a long time scale. Furthermore, we prove that with *any* constant stepsize, the GD iterates tend to infinity when projected to a max-margin direction (the hard-margin SVM direction) and converge to a fixed vector that minimizes a strongly convex potential when projected to the orthogonal complement of the max-margin direction. In contrast, we also show that in the EoS regime, GD iterates may diverge catastrophically under the exponential loss, highlighting the superiority of the logistic loss. These theoretical findings are in line with numerical simulations and complement existing theories on the convergence and implicit bias of GD for logistic regression, which are only applicable when the stepsizes are sufficiently small.

## 1 Introduction

*Gradient descent* (GD) is a foundational algorithm for machine learning optimization that motivates many popular algorithms. Theoretically, the behavior of GD is well understood when the stepsize is small. In this regard, one of the most classic results is the *descent lemma* (see, e.g., Section 1.2.3 in Nesterov et al. [2018]):

**Lemma** (Descent lemma, simplified version)**.** *Suppose that* $\sup_{\mathbf{w}} \left\| \nabla^2 L(\mathbf{w}) \right\|_2 \leq \beta$[1], *then*

$$L(\mathbf{w}_+) \leq L(\mathbf{w}) - \eta \cdot (1 - \eta\beta/2) \cdot \|\nabla L(\mathbf{w})\|_2^2, \quad \text{where } \mathbf{w}_+ := \mathbf{w} - \eta \cdot \nabla L(\mathbf{w}).$$

When the targeted function is smooth (such as logistic regression) and the stepsize is *small* ($0 < \eta < \beta/2$), the descent lemma ensures a monotonic decrease of the function value by performing each GD step. Building upon this, a sequence of iterates produced by GD with small stepsizes provably minimizes the function value in various settings (see, e.g., Lan [2020]).

For a more modern example, a recent line of research has established the *implicit bias* of GD with small stepsizes (see Soudry et al. [2018], Ji and Telgarsky [2018] and references thereafter). Specifically, they consider GD for optimizing logistic regression (besides other loss functions) on

---

[1]The uniformly bounded Hessian norm condition is stated for simplicity and can be relaxed in many ways. For example, it can be replaced by requiring $L(\cdot)$ to be $\beta$-smooth. For another example, it can also be replaced with $\sup_{0 \leq \lambda \leq 1} \left\| \nabla^2 L(\lambda \cdot \mathbf{w} + (1 - \lambda) \cdot \mathbf{w}_+) \right\|_2 \leq \beta$.

37th Conference on Neural Information Processing Systems (NeurIPS 2023).

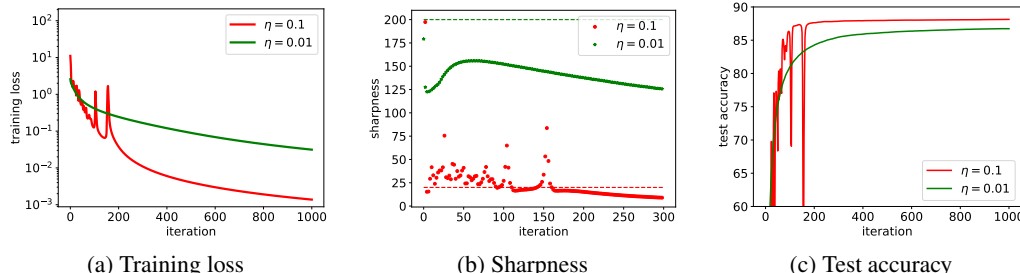

(a) Training loss      (b) Sharpness      (c) Test accuracy

Figure 1: The behaviors of GD for optimizing a neural network. We randomly sample $1,000$ data from the MNIST dataset and then use GD to train a $4$-layer fully connected network to fit those data. We use the cross-entropy loss, i.e., the multi-class version of the logistic loss. The sub-figures (a), (b), and (c) report the training loss, test accuracy, and sharpness (i.e., $\|\nabla L(\mathbf{w}_t)\|_2$) along the GD trajectories, respectively. The red curves correspond to GD with a large stepsize $\eta = 0.1$, where the training losses oscillate locally and the sharpnesses can exceed $2/\eta = 20$. The green curves correspond to GD with a small stepsize $\eta = 0.01$, where the training losses decrease monotonically and the sharpnesses are always below $2/\eta = 200$. Moreover, (c) suggests that large-stepsize GD achieves better test accuracy than small-stepsize GD, consistent with larger-scale deep learning experiments [Goyal et al., 2017]. More details of the experiments can be found in Appendix D.

linearly separable data. When the stepsizes are sufficiently small, the GD iterates are shown to decrease the risk monotonically (by a variant of the descent lemma); moreover, the GD iterates tend to align with a direction that maximizes the $\ell_2$-margin of the data [Soudry et al., 2018, Ji and Telgarsky, 2018]. The margin-maximization bias of small-stepsize GD sheds important light on understanding the statistical benefits of GD, as a large margin solution often generalizes well [Bartlett et al., 2017, Neyshabur et al., 2018].

Nonetheless, in practical machine learning optimization, especially in deep learning, the empirical risk (or training loss) often varies *non-monotonically* (while being minimized in the long run) — the local risk oscillation is not only caused by the algorithmic randomness but is more an effect of using *large stepsizes*, as it happens for deterministic GD (with large stepsizes) as well [Wu et al., 2018, Xing et al., 2018, Lewkowycz et al., 2020, Cohen et al., 2021]. This phenomenon is showcased in Figures 1(a) and 2(a), and is referred to by Cohen et al. [2021] as the *edge of stability* (EoS). The observation sets a non-negligible gap between practical and theoretical GD setups, where in practice, GD is run with large stepsizes that lead to local risk oscillations, but in theory, GD is only considered with sufficiently small stepsizes, predicting a monotonic risk descent (with a few exceptions, which will be discussed later in Section 2). A tension remains to be resolved:

> *Is the convergence of risk under local oscillation merely a "lucky" occurrence,*
> *or is it predictable based on theory?*

**Contributions.** In this work, we study the behaviors of GD in the EoS regime in arguably the simplest setting for machine learning optimization — logistic regression on linearly separable data. We show that with *any* constant stepsize, while the induced risks may oscillate locally, GD must minimize the risk in the long run at a rate of $\mathcal{O}(1/t)$, where $t$ is the number of iterates. In addition, we show that the direction of the GD iterates (with any constant stepsize) must align with a max-margin direction (the hard-margin SVM direction) at a rate of $\mathcal{O}(1/\log(t))$. These results explain how GD minimizes a risk non-monotonically, and complement existing theories [Soudry et al., 2018, Ji and Telgarsky, 2018] on the convergence and implicit bias of GD, which are only applicable when the stepsizes are sufficiently small.

Some additional notable contributions are

1. We also show that, when projected to the orthogonal complement of the max-margin direction, the GD iterates (with any constant stepsize) converge to a fixed vector that minimizes a strongly convex potential at a rate of $\mathcal{O}(1/\log(t))$. This characterization is conceptually more interpretable than an existing version [Soudry et al., 2018].

2. We show that in the EoS regime, GD can diverge catastrophically if the logistic loss is replaced by the exponential loss. This is in stark contrast to the small-stepsize regime, where the behaviors

of GD are known to be similar under any exponentially-tailed losses including both the logistic and exponential losses [Soudry et al., 2018, Ji and Telgarsky, 2018]. The difference in the EoS regime provides insights into why the logistic loss is preferable to the exponential loss in practice.

3. From a technical perspective, we develop a new approach for analyzing GD with large stepsizes. Our approach views the GD iterates as a coupling of two orthogonal iterates, one along a max-margin direction and the other along the orthogonal complement of the max-margin direction. The former iterates tend to infinity and the latter iterates approximate "imaginary" GD iterates that minimize a strongly convex potential with a *decaying* stepsize scheduler, controlled by the former iterates. Our techniques for analyzing large-stepsize GD can be of independent interest.

## 2 Related Works

In this section, we discuss papers related to our work.

**Implicit bias.** We first review a set of papers on the implicit bias of GD (with small stepsizes).

Along this line, Soudry et al. [2018] are the very first to show that GD converges along a max-margin direction when minimizing the empirical risk of an exponentially-tailed loss function (such as the logistic and exponential losses), a linear model, and linearly separable data. Then, an alternative analysis is provided by Ji and Telgarsky [2018], which also deals with non-separable data. These two works directly motivate us for considering GD for logistic regression on linearly separable data. However, there are at least three notable differences between our work and theirs. Firstly, their results only apply to GD with small stepsizes, while our results apply to GD with *any* constant stepsize. Secondly, their theories predict no difference between the logistic and exponential losses (as they are limited to the small-stepsize regime). Quite surprisingly, we prove that in the EoS regime, GD can diverge catastrophically under the exponential loss. Thirdly, from a technical viewpoint, their implicit bias analysis is built upon the risk convergence analysis, which further relies on a monotonic risk descent argument, hence only applies to small stepsizes. In comparison, we come up with a new approach that allows analyzing the implicit bias under risk oscillations; the long-term risk convergence is simply a consequence of the implicit bias results. Hence our techniques can accommodate any constant stepsize. See Section 5 for more discussions.

Subsequent works have extended the results by Soudry et al. [2018], Ji and Telgarsky [2018] to other algorithms such as momentum-based GD [Gunasekar et al., 2018a, Ji et al., 2021] and SGD [Nacson et al., 2019c], and homogenous but non-linear models [Gunasekar et al., 2017, Ji and Telgarsky, 2019, Gunasekar et al., 2018b, Nacson et al., 2019a, Lyu and Li, 2020] and non-homogenous models [Nacson et al., 2019a]. All these theories require the stepsizes to be small or even infinitesimal, in a regime away from our focus, the EoS regime.

It is worth noting that Nacson et al. [2019b] consider GD with an increasing stepsize scheduler that achieves a faster margin-maximization rate than constant-stepsize GD. However, their stepsize at each iteration is still appropriately small, resulting in a monotonic risk descent by a variant of the descent lemma.

**Edge of stability.** The risk oscillation phenomenon has been observed in several deep learning papers [Wu et al., 2018, Xing et al., 2018, Lewkowycz et al., 2020], and the work by Cohen et al. [2021] coins the term, *edge of stability* (EoS), that formally refers to it. In the remainder of this part, we focus on reviewing the current theoretical progress in understanding EoS.

Zhu et al. [2023] rigorously characterize EoS for a two-dimensional function $(u, v) \mapsto (u^2 v^2 - 1)^2$. Chen and Bruna [2022] study EoS for a one-dimensional function $u \mapsto (u^2 - 1)^2$ and for a special two-layer single-neuron network. Similar to these two works, Kreisler et al. [2023] study EoS in a 1-dimensional linear network. Ahn et al. [2022a] consider functions $(u, v) \mapsto \ell(uv)$, where $\ell$ is assumed to be convex, even, and Lipschitz; notably, they show a statistical gap between the small-stepsize regime and the EoS regime. Finally, Even et al. [2023], Andriushchenko et al. [2023] consider the regularization effect of large stepsizes in a diagonal linear network. Compared to their settings, our problem, i.e., logistic regression, is a natural machine-learning problem with fewer artifacts (if any).

EoS has also been theoretically investigated for general functions [Ma et al., 2022, Ahn et al., 2022b, Damian et al., 2022, Wang et al., 2022b], but these theories are often subject to subtle assumptions that are hard to interpret or verify. Specifically, Ma et al. [2022] require the function to grow

"subquadratically". Ahn et al. [2022b] assume the existence of a "forward invariant subset" near the set of minima of the function. Damian et al. [2022] assume a negative correlation between the gradient direction and the largest eigenvalue direction of the Hessian. Wang et al. [2022b] consider a two-layer neural network but require the norm of the last layer parameter and the sharpness to change in the same direction along the GD trajectory. Indirectly connected to EoS, the work by Kong and Tao [2020] shows a chaotic behavior of GD with a non-small stepsize when optimizing a "multi-scale" loss function. In comparison, our assumptions are more natural and interpretable.

Besides, the work by Lyu et al. [2022] considers EoS induced by GD for scale-invariant loss, e.g., a network with normalization layers and weight decay, and the work by Wang et al. [2022a] shows a balancing effect in matrix factorization induced by GD with a constant stepsize that is nearly $4/\|\nabla^2 L(\mathbf{w}_0)\|_2$ and is larger than $2/\|\nabla^2 L(\mathbf{w}_0)\|_2$. The objectives in their works are different from ours, i.e., logistic regression.

The unstable convergence has also been studied for normalized GD [Arora et al., 2022] and regularized GD [Bartlett et al., 2022]. These algorithms are apart from our focus on the vanilla GD.

Finally, the work by Liu et al. [2023] considers logistic regression with non-separable data (such that the objective is strongly convex), where GD with sufficiently large stepsize diverges. In contrast, we consider logistic regression with separable data, where GD with an arbitrarily large stepsize still converges.

## 3 Preliminaries

We use $\mathbf{x} \in \mathbb{R}^d$ to denote a feature vector and $y \in \{\pm 1\}$ to denote a binary label, respectively. Let $(\mathbf{x}_i, y_i)_{i=1}^n$ be a set of training data. Throughout the paper, we assume that $(\mathbf{x}_i, y_i)_{i=1}^n$ is *linearly separable* [Soudry et al., 2018].

**Assumption 1** (Linear separability). *Assume there is $\mathbf{w} \in \mathbb{R}^d$ such that $y_i \mathbf{x}_i^\top \mathbf{w} > 0$ for $i = 1, \ldots, n$.*

Let $\mathbf{w} \in \mathbb{R}^d$ be the parameter of a linear model. In *logistic regression*, we aim to minimize the following empirical risk

$$L(\mathbf{w}) := \sum_{i=1}^n \log \left( 1 + \exp(-y_i \cdot \langle \mathbf{x}_i, \mathbf{w} \rangle) \right), \quad \mathbf{w} \in \mathbb{R}^d.$$

We study a sequence of iterates $(\mathbf{w}_t)_{t \geq 0}$ produced by constant-stepsize *gradient descent* (GD), where $\mathbf{w}_0$ denotes the initialization and the remaining iterates are sequentially generated by:

$$\mathbf{w}_t = \mathbf{w}_{t-1} - \eta \cdot \nabla L(\mathbf{w}_{t-1}), \quad t \geq 1, \tag{GD}$$

where $\eta > 0$ is a constant stepsize. We are especially interested in a regime where $\eta$ is very large such that $L(\mathbf{w}_t)$ oscillates as a function of $t$. For the simplicity of presentation, we will assume that $\mathbf{w}_0 = 0$. Our results can be easily extended to allow general initialization.

The following notations are useful for presenting our results.

**Definition 1** (Margins and support vectors). Under Assumption 1, define the following notations:

(A) Let $\gamma$ be the max-$\ell_2$-margin (or max-margin in short), i.e.,

$$\gamma := \max_{\|\mathbf{w}\|_2 = 1} \min_{i \in [n]} y_i \cdot \langle \mathbf{x}_i, \mathbf{w} \rangle.$$

(B) Let $\hat{\mathbf{w}}$ be the hard-margin support-vector-machine (SVM) soluion, i.e.,

$$\hat{\mathbf{w}} := \arg \min_{\mathbf{w} \in \mathbb{R}^d} \|\mathbf{w}\|_2, \text{ s.t. } y_i \cdot \langle \mathbf{x}_i, \mathbf{w} \rangle \geq 1, \ i = 1, \ldots, n.$$

It is clear that $\hat{\mathbf{w}}$ exists and is uniquely defined (see, e.g., Section 5.2 in Mohri et al. [2018]). Note that $\|\hat{\mathbf{w}}\|_2 = 1/\gamma$ and $\hat{\mathbf{w}}/\|\hat{\mathbf{w}}\|_2$ is a max-margin direction. Also note that by duality, $\hat{\mathbf{w}}$ can be written as (see, e.g., Section 5.2 in Mohri et al. [2018])

$$\hat{\mathbf{w}} = \sum_{i \in \mathcal{S}} \alpha_i \cdot y_i \mathbf{x}_i, \quad \alpha_i \geq 0.$$

(C) Let $\mathcal{S}$ be the set of indexes of the support vectors, i.e.,

$$\mathcal{S} := \{i \in [n] : y_i \cdot \langle \mathbf{x}_i, \hat{\mathbf{w}}/\|\hat{\mathbf{w}}\|_2 \rangle = \gamma\}.$$

(D) If there exists non-support vector ($\mathcal{S} \subsetneq [n]$), let $\theta$ be the second smallest margin, i.e.,

$$\theta := \min_{i \notin \mathcal{S}} \; y_i \cdot \langle \mathbf{x}_i, \hat{\mathbf{w}}/\|\hat{\mathbf{w}}\|_2 \rangle.$$

It is clear from the definitions that $\theta > \gamma > 0$. In addition, from the definitions we have

$$\sum_{i \in \mathcal{S}} \alpha_i = \sum_{i \in \mathcal{S}} \alpha_i \cdot y_i \mathbf{x}_i^\top \hat{\mathbf{w}} = \|\hat{\mathbf{w}}\|_2^2 = \frac{1}{\gamma^2}.$$

In addition to Assumption 1, we make the following two mild assumptions to facilitate our analysis.

**Assumption 2** (Regularity conditions). *Assume that:*

*(A)* $\|\mathbf{x}_i\|_2 \le 1$, $i = 1, \ldots, n$.
*(B)* $\mathrm{rank}\{\mathbf{x}_i, \; i = 1, \ldots, n\} = d$.

Assumption 2 is only made for the convenience of presentation. In particular, Assumption 2(A) can be made true for any dataset by scaling the data vectors with a factor of $\max_i \|\mathbf{x}_i\|_2$. Without Assumption 2(B), our theorems still hold under a minor revision by replacing all the vectors of interests with their projections to $\mathrm{span}\{\mathbf{x}_i, i = 1, \ldots, n\}$.

**Assumption 3** (Non-degenerate data). *In addition to Assumption 1, assume that*

*(A)* $\mathrm{rank}\{\mathbf{x}_i, \; i \in \mathcal{S}\} = \mathrm{rank}\{\mathbf{x}_i, \; i = 1, \ldots, n\}$.
*(B)* *There exist* $\alpha_i > 0, i \in \mathcal{S}$ *such that* $\hat{\mathbf{w}} = \sum_{i \in \mathcal{S}} \alpha_i \cdot y_i \mathbf{x}_i$.

Assumption 3 has been used in Soudry et al. [2018] (see their Theorem 4), which requires that the support vectors span the dataset and are associated with strictly positive dual variables. Assumption 3(B) holds *almost surely* for every linearly separable dataset sampled from a continuous distribution according to Appendix B in Soudry et al. [2018]. Assumption 3 provides convenience to our analysis, but we conjecture it might not be necessary. Removing/relaxing Assumption 3 is left as a future work.

### 3.1 Space Decomposition

Conceptually, our analysis is built on a novel space decomposition viewpoint, which relies on the following lemma.

**Lemma 3.1** (Non-separable subspace). *Suppose that Assumptions 1, 2, and 3 hold. Then* $(\mathbf{x}_i, y_i)_{i \in \mathcal{S}}$ *is not linearly separable in the subspace orthogonal to the max-margin direction* $\hat{\mathbf{w}}/\|\hat{\mathbf{w}}\|_2$. *That is,*

*for every* $\mathbf{v}$ *such that* $\langle \mathbf{v}, \hat{\mathbf{w}} \rangle = 0$, *there exist* $i, j \in \mathcal{S}$ *such that* $y_i \cdot \langle \mathbf{x}_i, \mathbf{v} \rangle < 0$, $y_j \cdot \langle \mathbf{x}_j, \mathbf{v} \rangle > 0$.

*Proof of Lemma 3.1.* By Assumption 3 and $\langle \mathbf{v}, \hat{\mathbf{w}} \rangle = 0$, we have

$$0 = \langle \mathbf{v}, \hat{\mathbf{w}} \rangle = \sum_{i \in \mathcal{S}} \alpha_i \cdot y_i \mathbf{x}_i^\top \mathbf{v}.$$

By Assumptions 2 and 3 we have

$$\mathrm{rank}\{y_i \mathbf{x}_i, \; i \in \mathcal{S}\} = \mathrm{rank}\{\mathbf{x}_i, \; i \in \mathcal{S}\} = \mathrm{rank}\{\mathbf{x}_i, \; i = 1, \ldots, n\} = d,$$

so there must exist $i \in \mathcal{S}$ such that $y_i \mathbf{x}_i^\top \mathbf{v} \ne 0$. Without loss of generality, assume that $y_i \mathbf{x}_i^\top \mathbf{v} < 0$. Then since $\alpha_i > 0$ for $i \in \mathcal{S}$ by Assumption 3, there must exist $j \in \mathcal{S}$ such that $y_j \mathbf{x}_j^\top \mathbf{v} > 0$. $\qquad\square$

Lemma 3.1 shows that, although the dataset can be (linearly) separated by $\hat{\mathbf{w}}$, it cannot be separated by *any* vector orthogonal to $\hat{\mathbf{w}}$. This motivates us to decompose the $d$-dimensional ambient space into a 1-dimensional "separable" subspace and a $(d-1)$-dimensional "non-separable" subspace. This idea is formally realized as follows.

Fix $d-1$ orthogonal vectors $\mathbf{f}_1, \ldots, \mathbf{f}_{d-1} \in \mathbb{R}^d$ such that $\big(\hat{\mathbf{w}}/\|\hat{\mathbf{w}}\|_2, \mathbf{f}_1, \ldots, \mathbf{f}_{d-1}\big)$ forms an orthogonal basis of the ambient space $\mathbb{R}^d$. Then define two *projection operators*:

$$\mathcal{P}\colon \mathbb{R}^d \to \mathbb{R} \qquad \text{given by} \quad \mathbf{v} \mapsto \mathbf{v}^\top \hat{\mathbf{w}}/\|\hat{\mathbf{w}}\|_2,$$

$$\bar{\mathcal{P}}\colon \mathbb{R}^d \to \mathbb{R}^{d-1} \quad \text{given by} \quad \mathbf{v} \mapsto (\mathbf{v}^\top \mathbf{f}_1, \ldots, \mathbf{v}^\top \mathbf{f}_{d-1}).$$

The two operators together define a natural space decomposition, i.e., $\mathbb{R}^d = \mathcal{P}(\mathbb{R}^d) \oplus \bar{\mathcal{P}}(\mathbb{R}^d)$. Moreover, $\big(\mathcal{P}(\mathbf{x}_i), y_i\big)_{i=1}^n$ are linearly separable with an max-$\ell_2$-margin $\gamma$ according to Definition 1, and $\big(\bar{\mathcal{P}}(\mathbf{x}_i), y_i\big)_{i\in\mathcal{S}}$ (hence $\big(\bar{\mathcal{P}}(\mathbf{x}_i), y_i\big)_{i=1}^n$) are non-separable according to Lemma 3.1. So the decomposition of space can also be understood as the decomposition of data features into "max-margin features" and "non-separable features".

In what follows, we will call $\mathcal{P}(\mathbb{R}^d)$ the *max-margin subspace* and $\bar{\mathcal{P}}(\mathbb{R}^d)$ the *non-separable subspace*, respectively. In addition, we define a "margin offset" that quantifies to what extent the "non-separable features" are not separable.

**Definition 2** (Margin offset for the non-separable features)**.** Under Assumptions 1, 2, and 3, it holds that $\big(\bar{\mathcal{P}}(\mathbf{x}_i), y_i\big)_{i\in\mathcal{S}}$ is non-separable. Let $b$ be a *margin offset* such that

$$-b := \max_{\bar{\mathbf{w}}\in\mathbb{R}^{d-1},\, \|\bar{\mathbf{w}}\|_2=1} \ \min_{i\in\mathcal{S}} \ y_i \cdot \big\langle \bar{\mathcal{P}}(\mathbf{x}_i),\, \bar{\mathbf{w}} \big\rangle.$$

Then $b > 0$ due to the non-separability. The definition immediately implies that:

$$\text{for every } \bar{\mathbf{v}} \in \mathbb{R}^{d-1}, \text{ there exists } i \in \mathcal{S} \text{ such that } y_i \cdot \big\langle \bar{\mathcal{P}}(\mathbf{x}_i),\, \bar{\mathbf{v}} \big\rangle \leq -b \cdot \|\bar{\mathbf{v}}\|_2.$$

**Comparison to Ji and Telgarsky [2018].**   The work by Ji and Telgarsky [2018] also conducts space decomposition (see their Section 2). However, our approach is completely different from theirs. Firstly, they consider a non-separable dataset but we consider a linearly separable dataset. Secondly, at a higher level, they decompose the "dataset" (into two subsets), while we decompose the "features" (into two kinds of features). More specifically, Ji and Telgarsky [2018] first group the non-separable dataset into the "maximal linearly separable subset" and the complement, non-separable subset, then decompose the ambient space according to the subspace spanned by the non-separable subset and its orthogonal complement. In comparison, we consider a linearly separable dataset and decompose the ambient space according to a max-margin direction (i.e., $\mathcal{P}$) and its orthogonal complement (i.e., $\bar{\mathcal{P}}$).

# 4   Main Results

We are now ready to present our main results. All proofs are deferred to Appendix C. To begin with, we provide the following theorem that captures the behaviors of constant-stepsize GD for logistic regression on linearly separable data.

**Theorem 4.1** (The implicit bias of GD for logistic regression)**.** *Suppose that Assumptions 1, 2, and 3 hold. Consider $(\mathbf{w}_t)_{t\geq 0}$ produced by* (GD) *with initilization[2] $\mathbf{w}_0 = 0$ and constant stepsize $\eta > 0$. Then there exist positive constants $c_1, c_2, c_3 > 0$ that are upper bounded by a polynomial of $\big\{ e^\eta, e^n, e^{1/b}, 1/\eta, 1/(\theta-\gamma), 1/\gamma, e^{\theta/\gamma} \big\}$ but are independent of $t$, such that:*

*(A) The risk is upper bounded by*

$$L(\mathbf{w}_t) \leq c_1/t, \quad t \geq 3.$$

*(B) In the max-margin subspace,*

$$\mathcal{P}(\mathbf{w}_t) \geq \log(t)/\gamma + \log(\eta\gamma^2/2)/\gamma, \quad t \geq 1.$$

*(C) In the non-separable subspace,*

$$\big\|\bar{\mathcal{P}}(\mathbf{w}_t)\big\|_2 \leq c_2, \quad t \geq 0.$$

*(D) In addition, in the non-separable subspace,*

$$G\big(\bar{\mathcal{P}}(\mathbf{w}_t)\big) - \min G(\cdot) \leq c_3/\log(t), \quad t \geq 3,$$

*where $G(\cdot)$ is a strongly convex potential defined by*

$$G(\mathbf{v}) := \sum_{i\in\mathcal{S}} \exp\big( -y_i \cdot \big\langle \bar{\mathcal{P}}(\mathbf{x}_i),\, \mathbf{v} \big\rangle \big), \quad \mathbf{v} \in \mathbb{R}^{d-1}.$$

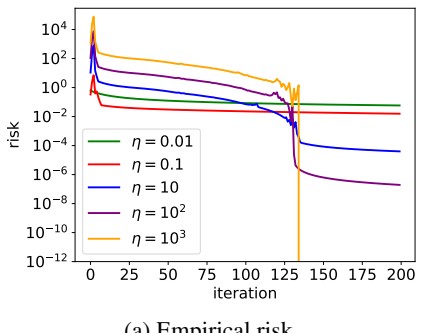
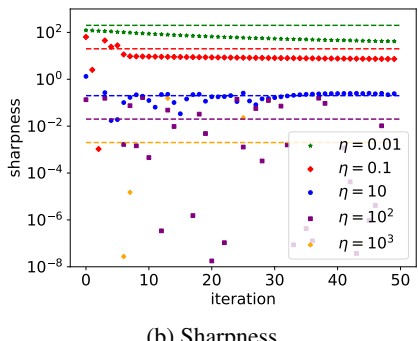

(a) Empirical risk

(b) Sharpness

Figure 2: The behaviors of GD for logistic regression. We randomly sample $1,000$ data with labels "0" and "8" from the MNIST dataset and then use GD to perform logistic regression on those data. The sub-figures (a) and (b) report the risk (i.e., the logistic loss) and sharpness (i.e., $\|\nabla L(\mathbf{w}_t)\|_2$) along the GD trajectories, respectively. For GD with stepsizes $\eta$ larger than or equal to $0.1$, the training losses oscillate locally and the sharpnesses can exceed $2/\eta$. For GD with a small stepsize $\eta = 0.01$, the training losses decrease monotonically and the sharpnesses are always below $2/\eta$. More details of the experiments can be found in Appendix D.

Note that Theorem 4.1 applies to GD with *any* positive constant stepsize, therefore allowing GD to be in the EoS regime. We next discuss the implications of Theorem 4.1 in detail.

**Risk minimization.** Theorem 4.1(A) guarantees that the GD iterates minimize the logistic loss at a rate of $\mathcal{O}(1/t)$ for any constant stepsize, even for those large stepsizes that cause local risk oscillations. This result explains the risk convergence of GD in the EoS regime, as illustrated in Figure 2, and is also consistent with the observations in neural network experiments (see Figure 1).

**Margin maximization.** Theorem 4.1(B) shows that the GD iterates, when projected to the max-margin direction, tend to infinity at a rate of $\mathcal{O}(\log(t))$. Moreover, Theorem 4.1(C) shows that the GD iterates, when projected to the non-separable subspace, are uniformly bounded. These two results together imply that the direction of the GD iterates will tend to a max-margin direction, i.e., the hard-margin SVM direction, at a rate of $\mathcal{O}(1/\log(t))$. Therefore, the implicit bias of GD that maximizes the $\ell_2$-margin is consistent in both the EoS regime and the small-stepsize regime [Soudry et al., 2018, Ji and Telgarsky, 2018].

**Iterate convergence in the non-separable subspace.** Theorem 4.1(D) shows that the GD iterates, when projected to the non-separable subspace, converge to the minimizer of a strongly convex potential $G(\cdot)$. Here, $G(\cdot)$ measures the exponential loss of a parameter on the support vectors with their non-separable features. This provides a more precise characterization of the implicit bias of GD: the direction of the GD iterates converges to the hard-margin SVM direction, moreover, the limit of the projections of the GD iterates to the orthogonal complement to the hard-margin SVM direction minimizes the exponential loss on the non-separable features of the support vectors.

**Comparison to Theorem 9 in Soudry et al. [2018].** Theorem 9, in particular, equation (18), in Soudry et al. [2018] *indirectly* characterizes the convergence of GD iterates in the non-separable subspace. It reads in our notations that: $\tilde{\mathbf{w}} := \lim_{t \to \infty} (\mathbf{w}_t - \hat{\mathbf{w}} \log(t))$ exists and satisfies

$$\text{for every } i \in \mathcal{S}, \ \ \eta \cdot \exp(-y_i \cdot \langle \mathbf{x}_i, \tilde{\mathbf{w}} \rangle) = \alpha_i, \text{ where } \alpha_i \text{ is defined in Assumption 3.} \tag{1}$$

In Appendix A, we show that Theorem 4.1(D) is equivalent to condition (1) in terms of describing $\bar{\mathcal{P}}(\mathbf{w}_\infty)$. Despite their equivalence, (1) is less interpretable than Theorem 4.1(D), as (1) entangles an effect of $\mathcal{P}(\mathbf{w}_\infty)$ with $\bar{\mathcal{P}}(\mathbf{w}_\infty)$, while Theorem 4.1 completely decouples $\mathcal{P}(\mathbf{w}_\infty)$ and $\bar{\mathcal{P}}(\mathbf{w}_\infty)$. In particular, (1) seems to suggest $\bar{\mathcal{P}}(\mathbf{w}_\infty)$ to be a function of stepsize $\eta$ since $\tilde{\mathbf{w}}$ depends on $\eta$. However, this is only an illusion brought by the lack of interpretability of (1); it is clear that $\bar{\mathcal{P}}(\mathbf{w}_\infty)$ is independent of $\eta$ according to Theorem 4.1(D).

**Exponential loss.** Until now, our theory for GD is consistent for large and small stepsizes. However, this is a particular benefit thanks to the design of the logistic loss, and may not hold for other losses. Our next result suggests that, in the EoS regime where the stepsizes are large, GD can diverge catastrophically under the exponential loss.

---

[2]The theorem can be easily extended to allow any $\mathbf{w}_0$ that has a bounded $\ell_2$-norm.

**Theorem 4.2** (The catastrophic divergence of GD under the exponential loss). *Consider a dataset of two samples, where*

$$\mathbf{x}_1 = (\gamma,\, 1), \quad y_1 = 1; \qquad \mathbf{x}_2 = (\gamma,\, -1), \quad y_2 = 1.$$

*It is clear that $(\mathbf{x}_i, y_i)_{i=1,2}$ is linearly separable and $(1, 0)$ is the max-margin direction. Consider a risk defined by the exponential loss:*

$$L(w, \bar{w}) := \exp(-y_1 \langle \mathbf{x}_1, \mathbf{w} \rangle) + \exp(-y_2 \langle \mathbf{x}_2, \mathbf{w} \rangle) = e^{-\gamma w} \cdot \left( e^{-\bar{w}} + e^{\bar{w}} \right), \quad \text{where } \mathbf{w} = (w, \bar{w}).$$

*Let $(w_t, \bar{w}_t)_{t \geq 0}$ be the iterates produced by GD with constant stepsize $\eta$ for optimizing $L(w, \bar{w})$. If*

$$0 \leq w_0 \leq 2, \quad |\bar{w}_0| \geq 1, \quad 0 < \gamma < 1/4, \quad \eta \geq 4,$$

*then:*

*(A) $L(w_t, \bar{w}_t) \to \infty$.*
*(B) $w_t \to \infty$.*
*(C) For every $t \geq 0$, $|\bar{w}_t| \geq 2\gamma w_t$.*
*(D) Moreover, the sign of $\bar{w}_t$ flips every iteration.*

*As a consequence, $(w_t, \bar{w}_t)_{t \geq 0}$ diverge in terms of either magnitude or direction; in particular, the direction of $(w_t, \bar{w}_t)_{t \geq 0}$ cannot converge to the max-margin direction (which is $(1, 0)$).*

Theorem 4.2 shows that with a large constant stepsize, the GD iterates no longer minimize the risk defined by the exponential loss and no longer converge along the max-margin direction. In fact, the directions of the GD iterates flip every step, thus the direction of the GD iterates necessarily *diverges*, resulting in no meaningful implicit bias at all.

In the EoS regime, large-stepsize GD still behaves nicely under the logistic loss (Theorem 4.1) but can behave catastrophically under the exponential loss (Theorem 4.2). From a mathematical standpoint, this difference is rooted in the fact that the gradient of the logistic loss is uniformly bounded while the gradient of the exponential loss could be extremely large. From a practical standpoint, it provides insights into why the logistics loss (and its multi-class version, the cross-entropy loss) is preferable to the exponential loss in practice.

The different behaviors of large-stepsize GD under the logistic and exponential losses also sharply contrast the EoS regime with the small-stepsize regime. Because in the small-stepsize regime, the convergence and implicit bias of GD are known to be similar under any exponentially-tailed losses, including the logistic and exponential losses [Soudry et al., 2018, Ji and Telgarsky, 2018].

## 5  Techniques Overview

The proofs of Theorems 4.1 and 4.2 are deferred to Appendix C. In this section, we explain the proof ideas of Theorem 4.1 by analyzing a simple dataset considered in Theorem 4.2 (the treatment to the general datasets can be found in Appendix B). But this time we work with the logistic loss instead of the exponential loss, that is,

$$L(w, \bar{w}) = \log(1 + e^{-\gamma w - \bar{w}}) + \log(1 + e^{-\gamma w + \bar{w}}).$$

Then the GD iterates can be written as

$$w_{t+1} = w_t - \eta \cdot g_t, \qquad \bar{w}_{t+1} = \bar{w}_t - \eta \cdot \bar{g}_t,$$

where

$$g_t := -\gamma \cdot \left( \frac{1}{1 + e^{\gamma w_t + \bar{w}_t}} + \frac{1}{1 + e^{\gamma w_t - \bar{w}_t}} \right), \quad \bar{g}_t := -\left( \frac{1}{1 + e^{\gamma w_t + \bar{w}_t}} - \frac{1}{1 + e^{\gamma w_t - \bar{w}_t}} \right).$$

For simplicity, assume that

$$w_0 = 0, \quad |\bar{w}_0| > 0.$$

Different from Soudry et al. [2018], Ji and Telgarsky [2018], our approach begins with showing the implicit bias (despite that the risk may oscillate). The long-term risk convergence is then simply a consequence of the implicit bias results.

**Step 1:** $(\bar{w}_t)_{t\geq 0}$ **is uniformly bounded.** Observe that $\bar{g}_t$ and $\bar{w}_t$ always share the same sign and that $|\bar{g}_t| \leq 1$, so we have

$$|\bar{w}_{t+1}| = \big||\bar{w}_t| - \eta \cdot |\bar{g}_t|\big| \leq \max\big\{|\bar{w}_t|,\ \eta \cdot |\bar{g}_t|\big\} \leq \max\big\{|\bar{w}_t|,\ \eta\big\}.$$

By induction, we get that $(|\bar{w}_t|)_{t\geq 0}$ is uniformly bounded by $\max\{|\bar{w}_0|,\ \eta\} = \Theta(1)$.

**Step 2:** $w_t \approx \log(t)/\gamma$**.** We turn to study the max-margin subspace. It is clear that $g_t \leq 0$ for every $t \geq 0$. So we have $w_t \geq 0$ by induction. Moreover, we have

$$-\frac{g_t}{\gamma} = \frac{e^{-\gamma w_t - \bar{w}_t}}{1 + e^{-\gamma w_t - \bar{w}_t}} + \frac{e^{-\gamma w_t + \bar{w}_t}}{1 + e^{-\gamma w_t + \bar{w}_t}} \leq e^{-\gamma w_t} \cdot e^{-\bar{w}_t} + e^{-\gamma w_t} \cdot e^{\bar{w}_t} \leq e^{-\gamma w_t} \cdot \Theta(1),$$

where the last inequality is because $|\bar{w}_t|$ is uniformly bounded. We also have

$$-\frac{g_t}{\gamma} = \frac{e^{-\gamma w_t - \bar{w}_t}}{1 + e^{-\gamma w_t - \bar{w}_t}} + \frac{e^{-\gamma w_t + \bar{w}_t}}{1 + e^{-\gamma w_t + \bar{w}_t}} \geq 0.5 \cdot \min\{1, e^{-\gamma w_t} e^{-\bar{w}_t}\} + 0.5 \cdot \min\{1, e^{-\gamma w_t} e^{\bar{w}_t}\}$$

$$\geq 0.5 \cdot \min\{1, e^{-\gamma w_t} e^{-\bar{w}_t} + e^{-\gamma w_t} e^{\bar{w}_t}\} \geq 0.5 \cdot \min\{1, e^{-\gamma w_t}\} = 0.5 \cdot e^{-\gamma w_t},$$

where the third inequality is because $e^{-\bar{w}_t} + e^{\bar{w}_t} \geq 1$ and the last equality is because $w_t \geq 0$. Putting these together, we have

$$g_t \approx -\gamma \cdot e^{-\gamma w_t} \cdot \Theta(1) \ \Rightarrow\ w_{t+1} \approx w_t - \eta\gamma \cdot e^{-\gamma w_t} \cdot \Theta(1) \ \Rightarrow\ w_t = \log(t)/\gamma \pm \Theta(1). \quad (2)$$

**Step 3:** $\bar{g}_t \approx \exp(-\gamma w_t) \cdot \nabla G(\bar{w}_t)$**.** We turn back to the non-separable subspace. Note that $\bar{g}_t$ is an odd function of $\bar{w}_t$. Without loss of generality, let us assume $\bar{w}_t \geq 0$ in this part. Notice that

$$\text{for every fixed } a > 1, \ f(t) := \frac{1}{t + 1/a} - \frac{1}{t + a} \text{ is a decreasing function of } t \geq 0. \quad (3)$$

Then we have

$$\bar{g}_t = e^{-\gamma w_t} \cdot \left(\frac{1}{e^{-\gamma w_t} + e^{-\bar{w}_t}} - \frac{1}{e^{-\gamma w_t} + e^{\bar{w}_t}}\right) \leq e^{-\gamma w_t} \cdot \left(\frac{1}{e^{-\bar{w}_t}} - \frac{1}{e^{\bar{w}_t}}\right) =: e^{-\gamma w_t} \cdot \nabla G(\bar{w}_t),$$

where the inequality is by (3), and $G(\bar{w}) := e^{\bar{w}} + e^{-\bar{w}}$ is defined as in Theorem 4.1(D). On the other hand, since $|\bar{w}_t|$ is bounded and $w_t$ is increasing (and tends to infinity), there must exist a time $t_0$ such that $e^{-\gamma w_t} \leq e^{-|\bar{w}_t|}$ for every $t \geq t_0$. Then for $t \geq t_0$ we have

$$\bar{g}_t = e^{-\gamma w_t} \cdot \left(\frac{1}{e^{-\gamma w_t} + e^{-\bar{w}_t}} - \frac{1}{e^{-\gamma w_t} + e^{\bar{w}_t}}\right) \geq e^{-\gamma w_t} \cdot \left(\frac{1}{2e^{-\bar{w}_t}} - \frac{1}{e^{-\bar{w}_t} + e^{\bar{w}_t}}\right)$$

$$= e^{-\gamma w_t} \cdot \frac{e^{\bar{w}_t} - e^{-\bar{w}_t}}{2e^{-2\bar{w}_t} + 2} \geq e^{-\gamma w_t} \cdot \frac{e^{\bar{w}_t} - e^{-\bar{w}_t}}{4} =: \frac{1}{4} \cdot e^{-\gamma w_t} \cdot \nabla G(\bar{w}_t),$$

where the first inequality is by (3) and $e^{-\gamma w_t} \leq e^{-\bar{w}_t}$, and the last inequality is because we assume $\bar{w}_t \geq 0$. Putting these together, and using (2), we obtain that

$$\text{for every } t \geq t_0, \ \bar{w}_{t+1} = \bar{w}_t - \eta_t \cdot \nabla G(\bar{w}_t), \text{ where } \eta_t \approx \eta \cdot e^{-\gamma w_t} \cdot \Theta(1) \approx \Theta(1)/t. \quad (4)$$

**Step 4: a modified descent lemma.** Using (4) and Taylor's expansion, we have

$$\text{for every } t \geq t_0, \ G(\bar{w}_{t+1}) \leq G(\bar{w}_t) - \eta_t \cdot \|\nabla G(\bar{w}_t)\|^2 + \frac{\beta}{2} \cdot \eta_t^2 \cdot \|\nabla G(\bar{w}_t)\|^2 \leq G(\bar{w}_t) + \frac{\Theta(1)}{t^2},$$

where $\beta := \sup_{|\bar{v}| \leq \max\{|\bar{w}_0|, \eta\}} \|\nabla^2 G(\bar{v})\|_2 = \Theta(1)$. Taking a telescoping sum from $t$ to $T$, we have

$$\text{for every } T \geq t \geq t_0, \ G(\bar{w}_T) \leq G(\bar{w}_t) + \Theta(1)/t. \quad (5)$$

**Step 5: the convergence of** $\bar{w}_t$**.** What remains is adapted from classic convergence arguments. Choose $\bar{w}_* = \arg\min G(\cdot)$, then

$$\|\bar{w}_{t+1} - \bar{w}_*\|_2^2 = \|\bar{w}_t - \bar{w}_*\|_2^2 - 2\eta_t \cdot \langle \bar{w}_t - \bar{w}_*, \nabla G(\bar{w}_t)\rangle + \eta_t^2 \cdot \|\nabla G(\bar{w}_t)\|_2^2$$

$$\leq \|\bar{w}_t - \bar{w}_*\|_2^2 - 2\eta_t \cdot (G(\bar{w}_t) - G(\bar{w}_*)) + \Theta(1)/t^2, \quad t \geq t_0,$$

where the equality is by (4), and the inequality is because of the convexity of $G(\cdot)$, $|\bar{w}_t| \le \Theta(1)$, and (4). Taking a telescoping sum, we have

$$\sum_{t=t_0}^{T} 2\eta_t \cdot (G(\bar{w}_t) - G(\bar{w}_*)) \le \|\bar{w}_{t_0} - \bar{w}_*\|_2^2 - \|\bar{w}_{T+1} - \bar{w}_*\|_2^2 + \sum_{t=t_0}^{T} \Theta(1)/t^2 \le \Theta(1).$$

Combing the above with (5) and using $\eta_t \approx \Theta(1)/t$ from (4), we get

$$\sum_{t=t_0}^{T} \eta_t \cdot (G(\bar{w}_T) - G(\bar{w}_*)) \le \sum_{t=t_0}^{T} \eta_t \cdot (G(\bar{w}_t) - G(\bar{w}_*)) + \sum_{t=t_0}^{T} \eta_t \cdot \Theta(1)/t \le \Theta(1).$$

Finally, since $\sum_{t=t_0}^{T} \eta_t \ge \Theta(1) \cdot (\log(T) - \log(t_0))$ according to (4), we get that $G(\bar{w}_T) - G(\bar{w}_*) \le \Theta(1)/(\log(T) - \log(t_0))$.

**Step 6: risk convergence.** The long-term risk convergence result can be easily established by making use of the implicit bias results we have obtained so far.

# 6   Conclusion

We consider constant-stepsize GD for logistic regression on linearly separable data. We show that with *any* constant stepsize, GD minimizes the logistic loss; moreover, the GD iterates tend to infinity when projected to a max-margin direction and tend to a fixed minimizer of a strongly convex potential when projected to the orthogonal complement of the max-margin direction. We also show that GD with a large stepsize may diverge catastrophically if the logistic loss is replaced by the exponential loss. Our theory explains how GD minimizes a risk non-monotonically.

## Acknolwdgement

We thank the anonymous reviewers for their helpful comments and Alexander Tsigler for pointing out several typos. VB is partially supported by the Ministry of Trade, Industry and Energy(MOTIE) and Korea Institute for Advancement of Technology (KIAT) through the International Cooperative R&D program. JDL acknowledges the support of the ARO under MURI Award W911NF-11-1-0304, the Sloan Research Fellowship, NSF CCF 2002272, NSF IIS 2107304, NSF CIF 2212262, ONR Young Investigator Award, and NSF CAREER Award 2144994.

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

# A    On the Equivalence between Theorem 4.1(D) and (1)

Note that $\tilde{\mathbf{w}}$ in (1) contains components in both the max-margin and non-separable subspaces, and we need to disentangle those two components.

Under the coordinate system that defines $\mathcal{P}$ and $\bar{\mathcal{P}}$, we can represent a vector $\mathbf{v} \in \mathbb{R}^d$ as

$$\mathbf{v} := \big(\mathcal{P}(\mathbf{v}), \bar{\mathcal{P}}(\mathbf{v})\big).$$

Then for $i \in \mathcal{S}$, we have

$$
\begin{aligned}
y_i \cdot \langle \mathbf{x}_i, \tilde{\mathbf{w}} \rangle &= y_i \cdot \big\langle \big(\mathcal{P}(\mathbf{x}_i), \bar{\mathcal{P}}(\mathbf{x}_i)\big),\ \big(\mathcal{P}(\tilde{\mathbf{w}}), \bar{\mathcal{P}}(\tilde{\mathbf{w}})\big)\big\rangle \\
&= y_i \cdot \mathcal{P}(\mathbf{x}_i) \cdot \mathcal{P}(\tilde{\mathbf{w}}) + y_i \cdot \big\langle \bar{\mathcal{P}}(\mathbf{x}_i),\ \bar{\mathcal{P}}(\tilde{\mathbf{w}})\big\rangle \qquad \text{since } \mathcal{P} \text{ and } \bar{\mathcal{P}} \text{ are orthogonal} \\
&= \gamma \mathcal{P}(\tilde{\mathbf{w}}) + y_i \cdot \big\langle \bar{\mathcal{P}}(\mathbf{x}_i),\ \bar{\mathcal{P}}(\tilde{\mathbf{w}})\big\rangle. \qquad \text{since } y_i \mathcal{P}(\mathbf{x}_i) = \gamma \text{ for } i \in \mathcal{S}
\end{aligned}
$$

So (1) is equivalent to

$$\text{for every } i \in \mathcal{S}, \quad \eta \exp\big(-\gamma \bar{\mathcal{P}}(\tilde{\mathbf{w}})\big) \cdot \exp\big(-y_i \cdot \big\langle \bar{\mathcal{P}}(\mathbf{x}_i),\ \bar{\mathcal{P}}(\tilde{\mathbf{w}})\big\rangle\big) = \alpha_i.$$

Recall that $\sum_{i \in \mathcal{S}} \alpha_i = 1/\gamma^2$ according to Definition 1. So focusing on $\bar{\mathcal{P}}$, the above is equivalent to the following condition on $\bar{\mathcal{P}}(\tilde{\mathbf{w}})$:

$$\alpha_i \propto \exp\big(-y_i \cdot \big\langle \bar{\mathcal{P}}(\mathbf{x}_i),\ \bar{\mathcal{P}}(\tilde{\mathbf{w}})\big\rangle\big), \quad i \in \mathcal{S}. \tag{6}$$

Here we ignore a shared normalization factor.

Now, recall from Assumption 3(B) that $(\alpha_i)_{i \in \mathcal{S}}$ are such that

$$\hat{\mathbf{w}} = \sum_{i \in \mathcal{S}} \alpha_i \cdot y_i \mathbf{x}_i, \quad \alpha_i > 0.$$

Note that as long as $\sum_{i \in \mathcal{S}} \alpha_i = 1/\gamma^2$, we have

$$\mathcal{P}(\hat{\mathbf{w}}) = \sum_{i \in \mathcal{S}} \alpha_i \cdot y_i \mathcal{P}(\mathbf{x}_i) = \gamma \cdot \frac{1}{\gamma^2} = \frac{1}{\gamma}.$$

Now consider $\bar{\mathcal{P}}$. Note that $\bar{\mathcal{P}}(\hat{\mathbf{w}}) = 0$ by the choice of $\bar{\mathcal{P}}$, then apply $\bar{\mathcal{P}}$ on both sides of the above equation, we get

$$0 = \bar{\mathcal{P}}(\hat{\mathbf{w}}) = \sum_{i \in \mathcal{S}} \alpha_i \cdot y_i \bar{\mathcal{P}}(\mathbf{x}_i). \tag{7}$$

Under (7), (6) is equivalent to the following condition on $\bar{\mathcal{P}}(\tilde{\mathbf{w}})$:

$$0 = \sum_{i \in \mathcal{S}} \exp\big(-y_i \cdot \big\langle \bar{\mathcal{P}}(\mathbf{x}_i), \bar{\mathcal{P}}(\tilde{\mathbf{w}})\big\rangle\big) \cdot y_i \bar{\mathcal{P}}(\mathbf{x}_i),$$

which is precisely the first-order condition of

$$\bar{\mathcal{P}}(\tilde{\mathbf{w}}) \in \arg\min G(\cdot), \text{ where } G(\mathbf{v}) := \sum_{i \in \mathcal{S}} \exp\big(-y_i \cdot \big\langle \bar{\mathcal{P}}(\mathbf{x}_i),\ \mathbf{v}\big\rangle\big).$$

Hence we have shown that: the condition that $\tilde{\mathbf{w}}$ satisfies (1) is equivalent to the condition that $\bar{\mathcal{P}}(\tilde{\mathbf{w}})$ minimizes the strongly convex potential $G(\cdot)$.

# B    The Behaviors of Constant-Stepsize GD

## B.1    Notation Simplifications

Without loss of generality, we assume that

$$y_i = 1, \quad i = 1, \ldots, n.$$

Otherwise, we replace $y_i$ with 1 and $\mathbf{x}_i$ with $y_i \cdot \mathbf{x}_i$, respectively, and the following analysis does not change.

Then the risk becomes

$$L(\mathbf{w}) := \sum_{i=1}^{n} \log(1 + e^{-\mathbf{w}^\top \mathbf{x}_i}).$$

**Rotating the hard-margin SVM solution.** Note that the (GD) iterates (under linear models) are rotation equivariant. Specifically, let $\mathbf{R}$ be an orthogonal matrix, then applying $\mathbf{R}$ on both sides of (GD), we obtain

$$\mathbf{R}\mathbf{w}_{t+1} = \mathbf{R}\mathbf{w}_t + \eta \sum_{i=1}^{n} \left(1 - s(\mathbf{x}_i^\top \mathbf{w}_t)\right) \cdot \mathbf{R}\mathbf{x}_i$$

$$= \mathbf{R}\mathbf{w}_t + \eta \sum_{i=1}^{n} \left(1 - s((\mathbf{R}\mathbf{x}_i)^\top (\mathbf{R}\mathbf{w}_t))\right) \cdot \mathbf{R}\mathbf{x}_i,$$

which is equivalent to the GD iterates under changes of variables, $\mathbf{w} \leftarrow \mathbf{R}\mathbf{w}$ and $\mathbf{x} \leftarrow \mathbf{R}\mathbf{x}$.

Therefore, without loss of generality, we can apply a rotation to the dataset such that $\hat{\mathbf{w}} \parallel \mathbf{e}_1$. Then for $\mathbf{v} \in \mathbb{R}^d$,

$$\mathcal{P}\mathbf{v} = \mathbf{v}[1] \in \mathbb{R}, \quad \bar{\mathcal{P}}\mathbf{v} = \mathbf{v}[2:d] \in \mathbb{R}^{d-1}.$$

Slightly abusing notations, in what follows we will write

$$\mathbf{x}_i = (x_i, \, \bar{\mathbf{x}}_i)^\top \in \mathbb{R} \oplus \mathbb{R}^{d-1}, \quad i = 1, \ldots, n,$$

where

$$x_i := \mathbf{x}_i[1] \in \mathbb{R}, \quad \bar{\mathbf{x}}_i := \mathbf{x}_i[2:d] \in \mathbb{R}^{d-1}.$$

Similarly, we define

$$\mathbf{w} = (w, \, \bar{\mathbf{w}})^\top \in \mathbb{R} \oplus \mathbb{R}^{d-1}.$$

Then we have

$$\mathbf{x}_i^\top \mathbf{w} = x_i w_i + \bar{\mathbf{x}}_i^\top \bar{\mathbf{w}}.$$

So the loss can be written as:

$$L(w, \bar{\mathbf{w}}) := \sum_{i=1}^{n} \log(1 + e^{-wx_i - \bar{\mathbf{w}}^\top \bar{\mathbf{x}}_i}).$$

So (GD) can be written as:

$$\begin{aligned} w_0 &= 0, \quad w_t = w_{t-1} - \eta \cdot \nabla_w L(w_{t-1}, \bar{\mathbf{w}}_{t-1}), \quad t \geq 1; \\ \bar{\mathbf{w}}_0 &= 0, \quad \bar{\mathbf{w}}_t = \bar{\mathbf{w}}_{t-1} - \eta \cdot \nabla_{\bar{\mathbf{w}}} L(w_{t-1}, \bar{\mathbf{w}}_{t-1}), \quad t \geq 1. \end{aligned} \tag{8}$$

The above two recursions capture the GD iterates projecting to the max-margin and non-separable subspaces, respectively.

## B.2 Boundedness of GD in the Non-Separable Subspace

We first show that $(\bar{\mathbf{w}}_t)_{t \geq 0}$ stay bounded for every fixed stepsize $\eta$.

**Lemma B.1** (Positiveness of $w_t$). *Suppose that Assumption1 holds. Consider $(w_t)_{t \geq 0}$ defined by (8) with constant stepsize $\eta > 0$. Then for every $t \geq 0$, it holds that $w_t \geq 0$.*

*Proof.* Recall that

$$w_0 = 0, \quad w_t = w_{t-1} - \eta \cdot \nabla_w L(w_{t-1}, \bar{\mathbf{w}}_{t-1}), \quad t \geq 1.$$

We only need to show that $\nabla_w L(w, \bar{\mathbf{w}}) \leq 0$. This is because

$$\nabla_w L(w, \bar{\mathbf{w}}) = -\sum_{i=1}^{n} \frac{1}{1 + e^{wx_i + \bar{\mathbf{w}}^\top \bar{\mathbf{x}}_i}} \cdot x_i$$

$$< 0. \qquad \qquad \text{since } x_i \geq \gamma > 0 \text{ by Definition 1}$$

$\square$

**Lemma B.2** (A recursion of $\|\bar{\mathbf{w}}_t\|_2$). *Suppose that Assumptions 1, 2, and 3 hold. Consider $(\bar{\mathbf{w}}_t)_{t\geq 0}$ defined by (8) with constant stepsize $\eta > 0$. Then for every $t \geq 0$, there exists $j \in [n]$ such that*

$$\|\bar{\mathbf{w}}_{t+1}\|_2^2 \leq \|\bar{\mathbf{w}}_t\|_2^2 + 2\eta e^{-w_t\gamma} \cdot \left(n - \frac{b \cdot \|\bar{\mathbf{w}}_t\|_2}{4}\right) + \frac{\eta}{1 + e^{w_t x_j + \bar{\mathbf{w}}_t^\top \bar{\mathbf{x}}_j}} \cdot \left(\eta n^2 - b \cdot \|\bar{\mathbf{w}}_t\|_2\right).$$

*As a direct consequence,*

$$\|\bar{\mathbf{w}}_t\|_2 \geq \max\{4n/b,\ \eta n^2/b\} \quad \textit{implies that} \quad \|\bar{\mathbf{w}}_{t+1}\|_2 \leq \|\bar{\mathbf{w}}_t\|_2.$$

*Proof.* We first make a few useful notations. Fix a time index $t$.

- Let $k$ be the index of the "most negatively classified" support sample, i.e.,

$$k := \arg\min_{i \in \mathcal{S}}\{\langle \bar{\mathbf{w}}_t, \bar{\mathbf{x}}_i\rangle\},$$

then by Definition 2 it holds that

$$\langle \bar{\mathbf{w}}_t, \bar{\mathbf{x}}_k\rangle \leq -b \cdot \|\bar{\mathbf{w}}_t\|_2. \tag{9}$$

- Let $j$ be the index of the "most negatively classified" sample, i.e.,

$$j := \arg\min_{1 \leq i \leq n}\{w_t x_i + \langle \bar{\mathbf{w}}_t, \bar{\mathbf{x}}_i\rangle\}.$$

Then

$$w_t x_j + \langle \bar{\mathbf{w}}_t, \bar{\mathbf{x}}_j\rangle \leq w_t x_i + \langle \bar{\mathbf{w}}_t, \bar{\mathbf{x}}_i\rangle \text{ for every } i \in [n]. \tag{10}$$

In particular, we must have

$$\langle \bar{\mathbf{w}}_t, \bar{\mathbf{x}}_j\rangle \leq -b\|\bar{\mathbf{w}}_t\|_2, \tag{11}$$

since

$$\begin{aligned}
w_t\gamma + \langle \bar{\mathbf{w}}_t, \bar{\mathbf{x}}_j\rangle &\leq w_t x_j + \langle \bar{\mathbf{w}}_t, \bar{\mathbf{x}}_j\rangle && \text{by Definition 1}\\
&\leq \min_{i \in \mathcal{S}}\{w_t x_i + \langle \bar{\mathbf{w}}_t, \bar{\mathbf{x}}_i\rangle\} && \text{by (10)}\\
&= w_t\gamma + \min_{i \in \mathcal{S}}\{\langle \bar{\mathbf{w}}_t, \bar{\mathbf{x}}_i\rangle\} && \text{by Definition 1}\\
&\leq w_t\gamma - b\|\bar{\mathbf{w}}_t\|_2. && \text{by Definition 2}
\end{aligned}$$

We remark that it is possible that $k = j$.

**Step 0: an iterate norm recursion.** Recall that

$$\bar{\mathbf{w}}_{t+1} = \bar{\mathbf{w}}_t - \eta\nabla_{\bar{\mathbf{w}}}L(w_t, \bar{\mathbf{w}}_t), \quad \nabla_{\bar{\mathbf{w}}}L(w_t, \bar{\mathbf{w}}_t) = -\sum_{i=1}^n \frac{1}{1 + e^{w_t x_i + \bar{\mathbf{w}}_t^\top \bar{\mathbf{x}}_i}} \cdot \bar{\mathbf{x}}_i.$$

Then

$$\|\bar{\mathbf{w}}_{t+1}\|_2^2 = \|\bar{\mathbf{w}}_t\|_2^2 - 2\eta \cdot \langle \bar{\mathbf{w}}_t, \nabla_{\bar{\mathbf{w}}}L(w_t, \bar{\mathbf{w}}_t)\rangle + \eta^2 \cdot \left\|\nabla_{\bar{\mathbf{w}}}L(w_t, \bar{\mathbf{w}}_t)\right\|_2^2.$$

**Step 1: gradient norm bounds.** By definition, we have

$$\begin{aligned}
\left\|\nabla_{\bar{\mathbf{w}}}L(w_t, \bar{\mathbf{w}}_t)\right\|_2 &= \left\|\sum_{i=1}^n \frac{1}{1 + e^{w_t x_i + \bar{\mathbf{w}}_t^\top \bar{\mathbf{x}}_i}} \cdot \bar{\mathbf{x}}_i\right\|_2\\
&\leq \sum_{i=1}^n \frac{1}{1 + e^{w_t x_i + \bar{\mathbf{w}}_t^\top \bar{\mathbf{x}}_i}} \cdot \|\bar{\mathbf{x}}_i\|_2\\
&\leq \sum_{i=1}^n \frac{1}{1 + e^{w_t x_i + \bar{\mathbf{w}}_t^\top \bar{\mathbf{x}}_i}} && \text{by Assumption 2}\\
&\leq \frac{n}{1 + e^{w_t x_j + \bar{\mathbf{w}}_t^\top \bar{\mathbf{x}}_j}} && \text{by (10)} \tag{12}
\end{aligned}$$

$$\leq n. \tag{13}$$

Here, we use a property of logistic loss that the gradient is uniformly bounded. Therefore, we have

$$\left\| \nabla_{\bar{\mathbf{w}}_t} L(w_t, \bar{\mathbf{w}}_t) \right\|_2^2 \leq \left( \frac{n}{1 + e^{w_t x_j + \bar{\mathbf{w}}_t^\top \bar{\mathbf{x}}_j}} \right) \cdot \left\| \nabla_{\bar{\mathbf{w}}_t} L(w_t, \bar{\mathbf{w}}_t) \right\|_2 \qquad \text{by (12)}$$

$$\leq \frac{n^2}{1 + e^{w_t x_j + \bar{\mathbf{w}}_t^\top \bar{\mathbf{x}}_j}}. \qquad \text{by (13)} \tag{14}$$

**Step 2: cross-term bounds.** We aim to show that the negative parts in the cross-term can cancel both the positve parts in the cross-term and the squared gradient norm term.

Note that the following holds for either $j = k$ or $j \neq k$:

$$- \langle \bar{\mathbf{w}}_t, \nabla_{\bar{\mathbf{w}}_t} L(w_t, \bar{\mathbf{w}}_t) \rangle$$

$$= \sum_{i=1}^n \frac{1}{1 + e^{w_t x_i + \bar{\mathbf{w}}_t^\top \bar{\mathbf{x}}_i}} \cdot \bar{\mathbf{w}}_t^\top \bar{\mathbf{x}}_i$$

$$\leq \sum_{\bar{\mathbf{w}}_t^\top \bar{\mathbf{x}}_i > 0} \frac{1}{1 + e^{w_t x_i + \bar{\mathbf{w}}_t^\top \bar{\mathbf{x}}_i}} \cdot \bar{\mathbf{w}}_t^\top \bar{\mathbf{x}}_i \tag{15}$$

$$+ \frac{1}{2} \cdot \frac{1}{1 + e^{w_t x_j + \bar{\mathbf{w}}_t^\top \bar{\mathbf{x}}_j}} \cdot \bar{\mathbf{w}}_t^\top \bar{\mathbf{x}}_j + \frac{1}{2} \cdot \frac{1}{1 + e^{w_t x_k + \bar{\mathbf{w}}_t^\top \bar{\mathbf{x}}_k}} \cdot \bar{\mathbf{w}}_t^\top \bar{\mathbf{x}}_k.$$

The first term in (15) can be bounded by

$$\sum_{\bar{\mathbf{w}}_t^\top \bar{\mathbf{x}}_i > 0} \frac{1}{1 + e^{w_t x_i + \bar{\mathbf{w}}_t^\top \bar{\mathbf{x}}_i}} \cdot \bar{\mathbf{w}}_t^\top \bar{\mathbf{x}}_i$$

$$= \sum_{\bar{\mathbf{w}}_t^\top \bar{\mathbf{x}}_i > 0} \frac{e^{-w_t x_i}}{1 + e^{-w_t x_i - \bar{\mathbf{w}}_t^\top \bar{\mathbf{x}}_i}} \cdot e^{-\bar{\mathbf{w}}_t^\top \bar{\mathbf{x}}_i} \cdot \bar{\mathbf{w}}_t^\top \bar{\mathbf{x}}_i$$

$$\leq \sum_{\bar{\mathbf{w}}_t^\top \bar{\mathbf{x}}_i > 0} \frac{e^{-w_t x_i}}{1 + e^{-w_t x_i - \bar{\mathbf{w}}_t^\top \bar{\mathbf{x}}_i}} \qquad \text{since } e^{-t} \cdot t \leq 1$$

$$\leq \sum_{\bar{\mathbf{w}}_t^\top \bar{\mathbf{x}}_i > 0} e^{-w_t x_i}$$

$$\leq n e^{-\gamma w_t}. \qquad \text{since } x_i \geq \gamma \text{ for } i \in [n] \tag{16}$$

The second term in (15) can be bounded by

$$\frac{1}{2} \cdot \frac{1}{1 + e^{w_t x_j + \bar{\mathbf{w}}_t^\top \bar{\mathbf{x}}_j}} \cdot \bar{\mathbf{w}}_t^\top \bar{\mathbf{x}}_j \leq \frac{1}{2} \cdot \frac{-b \cdot \|\bar{\mathbf{w}}_t\|_2}{1 + e^{w_t x_j + \bar{\mathbf{w}}_t^\top \bar{\mathbf{x}}_j}}. \qquad \text{by (11)} \tag{17}$$

The third term in (15) can be bounded by

$$\frac{1}{2} \cdot \frac{1}{1 + e^{w_t x_k + \bar{\mathbf{w}}_t^\top \bar{\mathbf{x}}_k}} \cdot \bar{\mathbf{w}}_t^\top \bar{\mathbf{x}}_k \leq \frac{1}{2} \cdot \frac{-b \cdot \|\bar{\mathbf{w}}_t\|_2}{1 + e^{w_t \gamma + \bar{\mathbf{w}}_t^\top \bar{\mathbf{x}}_k}} \qquad \text{by (9) and the choice of } k$$

$$= \frac{-b \cdot \|\bar{\mathbf{w}}_t\|_2}{2} \cdot \frac{e^{-w_t \gamma}}{e^{-w_t \gamma} + e^{\bar{\mathbf{w}}_t^\top \bar{\mathbf{x}}_k}}$$

$$\leq \frac{-b \cdot \|\bar{\mathbf{w}}_t\|_2}{2} \cdot \frac{e^{-w_t \gamma}}{2}, \qquad \text{since } e^{-w_t \gamma}, e^{\bar{\mathbf{w}}_t^\top \bar{\mathbf{x}}_k} \leq 1 \tag{18}$$

since

$$w_t \gamma \geq 0, \qquad \text{by Lemma B.1}$$

$$\bar{\mathbf{w}}_t^\top \bar{\mathbf{x}}_k \leq 0. \qquad \text{by the choice of } k$$

Now, bringing (16), (17), and (18) into (15), we obtain

$$-\langle \bar{\mathbf{w}}_t, \nabla_{\bar{\mathbf{w}}_t} L(w_t, \bar{\mathbf{w}}_t) \rangle \leq e^{-w_t \gamma} \cdot \left( n - \frac{b \cdot \|\bar{\mathbf{w}}_t\|_2}{4} \right) - \frac{b \cdot \|\bar{\mathbf{w}}_t\|_2}{2} \cdot \frac{1}{1 + e^{w_t x_j + \bar{\mathbf{w}}_t^\top \bar{\mathbf{x}}_j}}. \tag{19}$$

Here, we use a property of logistic loss that the gradients from incorrectly classified samples dominate the gradients from correctly classified samples.

**Step 3: iterate norm recursion bounds.** Using (14) and (19), we can obtain

$$
\begin{aligned}
\|\bar{\mathbf{w}}_{t+1}\|_2^2 &= \|\bar{\mathbf{w}}_t\|_2^2 - 2\eta \cdot \langle \bar{\mathbf{w}}_t, \nabla_{\bar{\mathbf{w}}} L(w_t, \bar{\mathbf{w}}_t) \rangle + \eta^2 \cdot \left\| \nabla_{\bar{\mathbf{w}}} L(w_t, \bar{\mathbf{w}}_t) \right\|_2^2 \\
&\leq \|\bar{\mathbf{w}}_t\|_2^2 + 2\eta e^{-w_t \gamma} \cdot \left( n - \frac{b \cdot \|\bar{\mathbf{w}}_t\|_2}{4} \right) - \eta b \cdot \|\bar{\mathbf{w}}_t\|_2 \cdot \frac{1}{1 + e^{w_t x_j + \bar{\mathbf{w}}_t^\top \bar{\mathbf{x}}_j}} \\
&\quad + \eta^2 \cdot \frac{n^2}{1 + e^{w_t x_j + \bar{\mathbf{w}}_t^\top \bar{\mathbf{x}}_j}} \\
&= \|\bar{\mathbf{w}}_t\|_2^2 + 2\eta e^{-w_t \gamma} \cdot \left( n - \frac{b \cdot \|\bar{\mathbf{w}}_t\|_2}{4} \right) + \frac{\eta}{1 + e^{w_t x_j + \bar{\mathbf{w}}_t^\top \bar{\mathbf{x}}_j}} \cdot \left( \eta n^2 - b \cdot \|\bar{\mathbf{w}}_t\|_2 \right).
\end{aligned}
$$

We have completed the proof. $\qquad \square$

**Lemma B.3** (Boundedness of $\bar{\mathbf{w}}$). *Suppose that Assumptions 1, 2, and 3 hold. Consider $(\bar{\mathbf{w}}_t)_{t \geq 0}$ defined by (8) with constant stepsize $\eta > 0$. Then for every $t \geq 0$, it holds that*

$$
\|\bar{\mathbf{w}}_t\|_2 \leq W_{\max} := \max\{4n/b, \eta n^2/b\} + \eta n.
$$

*Proof.* We prove the claim by induction. Clearly, $\|\bar{\mathbf{w}}_0\|_2 = 0 \leq \max\{4n/b, \eta n^2/b\} + \eta n$. Now suppose that

$$
\|\bar{\mathbf{w}}_t\|_2 \leq \max\{4n/b, \eta n^2/b\} + \eta n,
$$

and discuss the following two cases:

1. If $\|\bar{\mathbf{w}}_t\|_2 \leq \max\{4n/b, \eta n^2/b\}$, then

$$
\begin{aligned}
\|\bar{\mathbf{w}}_{t+1}\|_2 &\leq \|\bar{\mathbf{w}}_t\|_2 + \|\eta \cdot \nabla_{\bar{\mathbf{w}}} L(w_t, \bar{\mathbf{w}}_t)\|_2 && \text{by triangle inequality} \\
&\leq \|\bar{\mathbf{w}}_t\|_2 + \eta n && \text{by (13)} \\
&\leq \max\{4n/b, \eta n^2/b\} + \eta n.
\end{aligned}
$$

2. Else, we have

$$
\max\{4n/b, \eta n^2/b\} \leq \|\bar{\mathbf{w}}_t\|_2 \leq \max\{4n/b, \eta n^2/b\} + \eta n,
$$

which implies

$$
\begin{aligned}
\|\bar{\mathbf{w}}_{t+1}\|_2 &\leq \|\bar{\mathbf{w}}_t\|_2 && \text{by Lemma B.2} \\
&\leq \max\{4n/b, \eta n^2/b\} + \eta n.
\end{aligned}
$$

This completes the induction. $\qquad \square$

### B.3 Divergence of GD in the Max-Margin Subspace

**Definition 3** (Some loss measurements in the non-separable subspace). Under Assumptions 1, 2, and 3, we define the following notations:

(A) Define two loss functions

$$
G(\bar{\mathbf{w}}) := \sum_{i \in \mathcal{S}} e^{-\bar{\mathbf{w}}^\top \bar{\mathbf{x}}_i}, \qquad H(\bar{\mathbf{w}}) := \sum_{i \notin \mathcal{S}} e^{-\bar{\mathbf{w}}^\top \bar{\mathbf{x}}_i}.
$$

In the case where $\mathcal{S} = [n]$, we define $H(\bar{\mathbf{w}}) = 0$.

(B) Define

$$
G_{\min} := \min_{\bar{\mathbf{w}} \in \mathbb{R}^{d-1}} G(\bar{\mathbf{w}}),
$$

It is clear that $G_{\min} \geq 1$ since $(\bar{\mathbf{x}}_i)_{i \in \mathcal{S}}$ are non-separable by Definition 2.

(C) Define

$$
\bar{\mathbf{w}}_* := \arg \min_{\bar{\mathbf{w}} \in \mathbb{R}^{d-1}} G(\bar{\mathbf{w}}).
$$

It is clear that $G(\bar{\mathbf{w}}_*) = G_{\min}$. Moreover, it holds that $\|\bar{\mathbf{w}}_*\|_2 \leq W_{\max}$ by Lemma B.4.

(D) Recall that $\|\bar{\mathbf{w}}_t\|_2 \leq W_{\max}$ according to Lemma B.3. We then define

$$G_{\max} := \sup_{\|\bar{\mathbf{w}}\|_2 \leq W_{\max}} G(\bar{\mathbf{w}}), \qquad H_{\max} := \sup_{\|\bar{\mathbf{w}}\|_2 \leq W_{\max}} H(\bar{\mathbf{w}}).$$

It is clear that

$$G(\bar{\mathbf{w}}_t) \leq G_{\max}, \qquad H(\bar{\mathbf{w}}_t) \leq H_{\max},$$

and that $G_{\max}$, $H_{\max}$ are polynomials on $e^\eta$, $e^n$, and $e^{1/b}$, and are independent of $t$.

**Lemma B.4.** *For the $\bar{\mathbf{w}}_*$ in Definition 3, it holds that*

$$\|\bar{\mathbf{w}}_*\|_2 \leq \frac{\log(n)}{b} \leq W_{\max}.$$

*Proof.* By Definition 2, there exists $j \in \mathcal{S}$ such that

$$\bar{\mathbf{w}}_*^\top \bar{\mathbf{x}}_j \leq -b \cdot \|\bar{\mathbf{w}}_*\|_2,$$

which implies that

$$G(\bar{\mathbf{w}}_*) = \sum_{i \in \mathcal{S}} e^{-\bar{\mathbf{w}}_*^\top \bar{\mathbf{x}}_i} \geq e^{-\bar{\mathbf{w}}_*^\top \bar{\mathbf{x}}_j} \geq e^{b \cdot \|\bar{\mathbf{w}}_*\|_2}.$$

On the other hand, by the definition of $\bar{\mathbf{w}}_*$, we have

$$G(\bar{\mathbf{w}}_*) \leq G(0) = n.$$

Therefore, we have $e^{b \cdot \|\bar{\mathbf{w}}_*\|_2} \leq n$, that is, $\|\bar{\mathbf{w}}_*\|_2 \leq \log(n)/b \leq W_{\max}$. $\qquad \square$

We now consider $(w_t)_{t \geq 0}$.

**Lemma B.5.** *Suppose Assumptions 1, 2, and 3 hold. Then for every $t \geq 0$, it holds that*

$$w_{t+1} \geq w_t + \frac{\eta\gamma}{2} \cdot \min\left\{1, \ e^{-\gamma w_t} \cdot G(\bar{\mathbf{w}}_t)\right\},$$

$$w_{t+1} \leq w_t + \eta \cdot \min\left\{\gamma n, \ \gamma \cdot e^{-\gamma w_t} \cdot G(\bar{\mathbf{w}}_t) + \eta \cdot e^{-\theta w_t} \cdot H(\bar{\mathbf{w}}_t)\right\}.$$

*Proof.* Recall that

$$w_{t+1} = w_t - \eta \cdot \nabla_w L(w_t, \bar{\mathbf{w}}_t), \qquad \nabla_w L(w_t, \bar{\mathbf{w}}_t) = -\sum_{i=1}^n \frac{e^{-x_i w_t - \bar{\mathbf{x}}_i^\top \bar{\mathbf{w}}_t}}{1 + e^{-x_i w_t - \bar{\mathbf{x}}_i^\top \bar{\mathbf{w}}_t}} \cdot x_i.$$

We only need to provide upper and lower bounds on $-\nabla_w L(w_t, \bar{\mathbf{w}}_t)$. The lower bound is because:

$$-\nabla_w L(w_t, \bar{\mathbf{w}}_t) = \sum_{i=1}^n \frac{e^{-x_i w_t - \bar{\mathbf{x}}_i^\top \bar{\mathbf{w}}_t}}{1 + e^{-x_i w_t - \bar{\mathbf{x}}_i^\top \bar{\mathbf{w}}_t}} \cdot x_i$$

$$\geq \sum_{i \in \mathcal{S}} \frac{e^{-x_i w_t - \bar{\mathbf{x}}_i^\top \bar{\mathbf{w}}_t}}{1 + e^{-x_i w_t - \bar{\mathbf{x}}_i^\top \bar{\mathbf{w}}_t}} \cdot x_i \qquad \text{since } x_i \geq \gamma > 0 \text{ by Definition 1}$$

$$= \sum_{i \in \mathcal{S}} \frac{e^{-\gamma w_t - \bar{\mathbf{x}}_i^\top \bar{\mathbf{w}}_t}}{1 + e^{-\gamma w_t - \bar{\mathbf{x}}_i^\top \bar{\mathbf{w}}_t}} \cdot \gamma \qquad \text{since } x_i = \gamma \text{ for } i \in \mathcal{S}$$

$$\geq \frac{\gamma}{2} \cdot \sum_{i \in \mathcal{S}} \min\{1, \ e^{-\gamma w_t - \bar{\mathbf{x}}_i^\top \bar{\mathbf{w}}_t}\} \qquad \text{since } e^t/(1 + e^t) \geq 0.5 \min\{1, e^t\}$$

$$\geq \frac{\gamma}{2} \cdot \min\left\{1, \ e^{-\gamma w_t} \cdot \sum_{i \in \mathcal{S}} e^{-\bar{\mathbf{x}}_i^\top \bar{\mathbf{w}}_t}\right\}$$

$$= \frac{\gamma}{2} \cdot \min\left\{1, e^{-\gamma w_t} \cdot G(\bar{\mathbf{w}}_t)\right\}.$$

The upper bound is because:

$$-\nabla_w L(w_t, \bar{\mathbf{w}}_t) = \sum_{i=1}^n \frac{e^{-x_i w_t - \bar{\mathbf{x}}_i^\top \bar{\mathbf{w}}_t}}{1 + e^{-x_i w_t - \bar{\mathbf{x}}_i^\top \bar{\mathbf{w}}_t}} \cdot x_i$$

$$
\begin{aligned}
&= \sum_{i \in \mathcal{S}} \frac{e^{-x_i w_t - \bar{\mathbf{x}}_i^\top \bar{\mathbf{w}}_t}}{1 + e^{-x_i w_t - \bar{\mathbf{x}}_i^\top \bar{\mathbf{w}}_t}} \cdot x_i + \sum_{i \notin \mathcal{S}} \frac{e^{-x_i w_t - \bar{\mathbf{x}}_i^\top \bar{\mathbf{w}}_t}}{1 + e^{-x_i w_t - \bar{\mathbf{x}}_i^\top \bar{\mathbf{w}}_t}} \cdot x_i \\
&\leq \sum_{i \in \mathcal{S}} \frac{e^{-\gamma w_t - \bar{\mathbf{x}}_i^\top \bar{\mathbf{w}}_t}}{1 + e^{-\gamma w_t - \bar{\mathbf{x}}_i^\top \bar{\mathbf{w}}_t}} \cdot \gamma + \sum_{i \notin \mathcal{S}} \frac{e^{-x_i w_t - \bar{\mathbf{x}}_i^\top \bar{\mathbf{w}}_t}}{1 + e^{-x_i w_t - \bar{\mathbf{x}}_i^\top \bar{\mathbf{w}}_t}} \\
&\qquad \text{since } x_i = \gamma \text{ for } i \in \mathcal{S}, \text{ and } x_i \leq 1 \text{ for } i \in [n] \\
&\leq \gamma \cdot \sum_{i \in \mathcal{S}} \min\{1,\, e^{-\gamma w_t - \bar{\mathbf{x}}_i^\top \bar{\mathbf{w}}_t}\} + \sum_{i \notin \mathcal{S}} \min\{1,\, e^{-w_t x_i - \bar{\mathbf{x}}_i^\top \bar{\mathbf{w}}_t}\} \\
&\qquad \text{since } e^t/(1 + e^t) \leq \min\{1, e^t\} \\
&\leq \gamma \cdot \sum_{i \in \mathcal{S}} \min\left\{1,\, e^{-\gamma w_t - \bar{\mathbf{x}}_i^\top \bar{\mathbf{w}}_t}\right\} + \sum_{i \notin \mathcal{S}} \min\left\{1,\, e^{-\theta w_t - \bar{\mathbf{x}}_i^\top \bar{\mathbf{w}}_t}\right\} \\
&\qquad \text{since } x_t \geq \theta > \gamma \text{ for } i \notin \mathcal{S} \\
&\leq \gamma \cdot \sum_{i \in \mathcal{S}} e^{-\gamma w_t - \bar{\mathbf{x}}_i^\top \bar{\mathbf{w}}_t} + \sum_{i \notin \mathcal{S}} e^{-\theta w_t - \bar{\mathbf{x}}_i^\top \bar{\mathbf{w}}_t} \\
&= \gamma e^{-\gamma w_t} \cdot G(\bar{\mathbf{w}}_t) + e^{-\theta w_t} \cdot H(\bar{\mathbf{w}}_t).
\end{aligned}
$$

We have completed the proof. $\qquad\square$

**Lemma B.6** (A lower bound on $w_t$). *Suppose Assumptions 2, 1, and 3 hold. Then it holds that*

$$
w_t \geq \frac{1}{\gamma} \cdot \log\left(1 + \frac{\eta \gamma^2}{2} \cdot t\right), \quad t \geq 0.
$$

*As a direct consequence, it holds that*

$$
e^{-\gamma w_t} \leq \frac{2}{2 + \eta \gamma^2 \cdot t}, \quad t \geq 0.
$$

*Proof.* Observe that

$$
\begin{aligned}
w_{t+1} &\geq w_t + \frac{\eta \gamma}{2} \cdot \min\left\{1, e^{-\gamma w_t} \cdot G(\bar{\mathbf{w}}_t)\right\} && \text{by Lemma B.5} \\
&\geq w_t + \frac{\eta \gamma}{2} \cdot \min\left\{1, e^{-\gamma w_t} \cdot 1\right\} && \text{by Definition 3} \\
&\geq w_t + \frac{\eta \gamma}{2} \cdot e^{-\gamma w_t}, && \text{since } w_t \geq 0 \text{ by Lemma B.1} \quad (20)
\end{aligned}
$$

which implies that $w_t$ is increasing. Furthermore, we have

$$
\begin{aligned}
e^{\gamma w_{t+1}} - e^{\gamma w_t} &= e^{\gamma w_t} \cdot \left(e^{\gamma(w_{t+1} - w_t)} - 1\right) \\
&\geq e^{\gamma w_t} \cdot \gamma(w_{t+1} - w_t) && \text{since } e^t - 1 \geq t \text{ for } t \geq 0, \text{ and } w_{t+1} \geq w_t \\
&\geq \frac{\eta \gamma^2}{2}, && \text{by (20)}
\end{aligned}
$$

which implies that

$$
\begin{aligned}
e^{\gamma w_t} &\geq e^{\gamma w_0} + \frac{\eta \gamma^2}{2} \cdot t \\
&= 1 + \frac{\eta \gamma^2}{2} \cdot t. && \text{since } w_0 = 0
\end{aligned}
$$

We then get

$$
w_t \geq \frac{1}{\gamma} \cdot \log\left(1 + \frac{\eta \gamma^2}{2} \cdot t\right), \quad t \geq 0.
$$

$\qquad\square$

**Lemma B.7** (An upper bound on $w_t$). *Suppose Assumptions 1, 2, and 3 hold. Then it holds that*

$$w_t \leq \frac{1}{\gamma} \cdot \log\left(\left(e\eta\gamma^2 G_{\max} + e\eta\gamma H_{\max}\right) \cdot (t+1)\right), \quad t \geq 0.$$

*As a direct consequence, it holds that*

$$e^{-\gamma w_t} \geq \frac{1}{\left(e\eta\gamma^2 G_{\max} + e\eta\gamma H_{\max}\right) \cdot (t+1)}, \quad t \geq 0.$$

*Proof.* Observe that

$$
\begin{aligned}
w_{t+1} - w_t &\leq \eta\gamma \cdot e^{-\gamma w_t} \cdot G(\bar{\mathbf{w}}_t) + \eta \cdot e^{-\theta w_t} \cdot H(\bar{\mathbf{w}}_t) && \text{by Lemma B.5} \\
&\leq \eta\gamma \cdot e^{-\gamma w_t} \cdot G(\bar{\mathbf{w}}_t) + \eta \cdot e^{-\gamma w_t} \cdot H(\bar{\mathbf{w}}_t) && \text{since } \theta > \gamma \text{ by Definition 1} \\
&\leq \eta \cdot \left(\gamma G_{\max} + H_{\max}\right) \cdot e^{-\gamma w_t} && \text{by Definition 3} \quad (21)
\end{aligned}
$$

Let

$$t_0 := \min\left\{ t : \gamma\eta \cdot \left(\gamma G_{\max} + H_{\max}\right) \cdot e^{-\gamma w_t} \leq 1 \right\}.$$

Recall that $w_t$ is increasing according to (20). So we have

$$\text{for } t \leq t_0, \qquad w_t \leq \frac{1}{\gamma} \cdot \log\left(\eta\gamma^2 G_{\max} + \eta\gamma H_{\max}\right); \tag{22}$$

$$\text{for } t \geq t_0, \qquad \gamma\eta \cdot \left(\gamma G_{\max} + H_{\max}\right) \cdot e^{-\gamma w_t} \leq 1. \tag{23}$$

(21) and (23) together imply that

$$\text{for } t \geq t_0, \qquad 0 \leq \gamma \cdot \left(w_{t+1} - w_t\right) \leq 1. \tag{24}$$

Then for $t \geq t_0$, we have

$$
\begin{aligned}
e^{\gamma w_{t+1}} - e^{\gamma w_t} &= e^{\gamma w_t}\left(e^{\gamma(w_{t+1} - w_t)} - 1\right) \\
&\leq e^{\gamma w_t} \cdot e \cdot \gamma(w_{t+1} - w_t) && \text{by (24) and that } e^t - 1 \leq e \cdot t \text{ for } 0 \leq t \leq 1 \\
&\leq e\eta\gamma^2 G_{\max} + e\eta\gamma H_{\max}, && \text{by (21).}
\end{aligned}
$$

which implies

$$
\begin{aligned}
e^{\gamma w_t} &\leq e^{\gamma w_{t_0}} + \left(e\eta\gamma^2 G_{\max} + e\eta\gamma H_{\max}\right) \cdot (t - t_0) \\
&\leq \eta\gamma^2 G_{\max} + \eta\gamma H_{\max} + \left(e\eta\gamma^2 G_{\max} + e\eta\gamma H_{\max}\right) \cdot (t - t_0) && \text{by (22)} \\
&\leq \left(e\eta\gamma^2 G_{\max} + e\eta\gamma H_{\max}\right) \cdot (t + 1).
\end{aligned}
$$

Therefore, for $t \geq t_0$, we have

$$w_t \leq \frac{1}{\gamma} \cdot \log\left(\left(e\eta\gamma^2 G_{\max} + e\eta\gamma H_{\max}\right) \cdot (t+1)\right).$$

Note that the above also holds for $0 \leq t \leq t_0$ according to (22). We have completed the proof. $\quad\square$

### B.4 Convergence of GD in the Non-Separable Subspace

We show that the vanilla gradient on the non-separable subspace, $\nabla_{\bar{\mathbf{w}}} L(w_t, \bar{\mathbf{w}}_t)$, can be understood as the gradient on a modified loss with a rescaling factor, $e^{-\gamma w_t} \nabla G(\bar{\mathbf{w}}_t)$, ignoring higher order errors.

**Lemma B.8** (Gradients comparison lemma). *Suppose Assumptions 1, 2, and 3 hold. Then it holds that*

$$\left\|\nabla_{\bar{\mathbf{w}}} L(w_t, \bar{\mathbf{w}}_t) - e^{-\gamma w_t} \cdot \nabla G(\bar{\mathbf{w}}_t)\right\|_2 \leq e^{-2\gamma w_t} \cdot G_{\max}^2 + e^{-\theta w_t} \cdot H_{\max}, \quad t \geq 0.$$

*As a direct consequence, for every vector $\bar{\mathbf{v}} \in \mathbb{R}^{d-1}$, it holds that*

$$\langle \bar{\mathbf{v}}, \, \nabla_{\bar{\mathbf{w}}} L(w_t, \bar{\mathbf{w}}_t) \rangle \leq e^{-\gamma w_t} \cdot \langle \bar{\mathbf{v}}, \, \nabla G(\bar{\mathbf{w}}_t) \rangle + \|\bar{\mathbf{v}}\|_2 \cdot \left(e^{-2\gamma w_t} \cdot G_{\max}^2 + e^{-\theta w_t} \cdot H_{\max}\right).$$

*Proof.* Recall that

$$\nabla_{\bar{\mathbf{w}}}L(w_t,\bar{\mathbf{w}}_t) = -\sum_{i=1}^{n}\frac{e^{-w_t x_i - \bar{\mathbf{w}}_t^\top \bar{\mathbf{x}}_i}}{1 + e^{-w_t x_i - \bar{\mathbf{w}}_t^\top \bar{\mathbf{x}}_i}}\cdot\bar{\mathbf{x}}_i \qquad \nabla G(\bar{\mathbf{w}}_t) = -\sum_{i\in\mathcal{S}}e^{-\bar{\mathbf{w}}_t^\top \bar{\mathbf{x}}_i}\cdot\bar{\mathbf{x}}_i.$$

By the triangle inequality, we have

$$\left\|\nabla_{\bar{\mathbf{w}}}L(w_t,\bar{\mathbf{w}}_t) - e^{-\gamma w_t}\nabla G(\bar{\mathbf{w}}_t)\right\|_2$$

$$= \left\|\sum_{i\in\mathcal{S}}\left(e^{-\gamma w_t - \bar{\mathbf{w}}_t^\top \bar{\mathbf{x}}_i} - \frac{e^{-w_t x_i - \bar{\mathbf{w}}_t^\top \bar{\mathbf{x}}_i}}{1 + e^{-w_t x_i - \bar{\mathbf{w}}_t^\top \bar{\mathbf{x}}_i}}\right)\cdot\bar{\mathbf{x}}_i - \sum_{i\notin\mathcal{S}}\frac{e^{-w_t x_i - \bar{\mathbf{w}}_t^\top \bar{\mathbf{x}}_i}}{1 + e^{-w_t x_i - \bar{\mathbf{w}}_t^\top \bar{\mathbf{x}}_i}}\cdot\bar{\mathbf{x}}_i\right\|_2$$

$$\leq \underbrace{\left\|\sum_{i\in\mathcal{S}}\left(e^{-\gamma w_t - \bar{\mathbf{w}}_t^\top \bar{\mathbf{x}}_i} - \frac{e^{-w_t x_i - \bar{\mathbf{w}}_t^\top \bar{\mathbf{x}}_i}}{1 + e^{-w_t x_i - \bar{\mathbf{w}}_t^\top \bar{\mathbf{x}}_i}}\right)\cdot\bar{\mathbf{x}}_i\right\|_2}_{(\clubsuit)} + \underbrace{\left\|\sum_{i\notin\mathcal{S}}\frac{e^{-w_t x_i - \bar{\mathbf{w}}_t^\top \bar{\mathbf{x}}_i}}{1 + e^{-w_t x_i - \bar{\mathbf{w}}_t^\top \bar{\mathbf{x}}_i}}\cdot\bar{\mathbf{x}}_i\right\|_2}_{(\heartsuit)}. \qquad (25)$$

The ($\clubsuit$) term can be bounded by

$$(\clubsuit) = \left\|\sum_{i\in\mathcal{S}}\left(e^{-\gamma w_t - \bar{\mathbf{w}}_t^\top \bar{\mathbf{x}}_i} - \frac{e^{-\gamma w_t - \bar{\mathbf{w}}_t^\top \bar{\mathbf{x}}_i}}{1 + e^{-\gamma w_t - \bar{\mathbf{w}}_t^\top \bar{\mathbf{x}}_i}}\right)\cdot\bar{\mathbf{x}}_i\right\|_2 \qquad \text{since } x_i = \gamma \text{ for } i\in\mathcal{S}$$

$$= \left\|\sum_{i\in\mathcal{S}}\frac{e^{-\gamma w_t - \bar{\mathbf{w}}_t^\top \bar{\mathbf{x}}_i}}{1 + e^{-\gamma w_t - \bar{\mathbf{w}}_t^\top \bar{\mathbf{x}}_i}}\cdot e^{-\gamma w_t - \bar{\mathbf{w}}_t^\top \bar{\mathbf{x}}_i}\cdot\bar{\mathbf{x}}_i\right\|_2$$

$$= e^{-2\gamma w_t}\cdot\left\|\sum_{i\in\mathcal{S}}\frac{1}{1 + e^{-\gamma w_t - \bar{\mathbf{w}}_t^\top \bar{\mathbf{x}}_i}}\cdot e^{-2\bar{\mathbf{w}}_t^\top \bar{\mathbf{x}}_i}\cdot\bar{\mathbf{x}}_i\right\|_2$$

$$\leq e^{-2\gamma w_t}\cdot\sum_{i\in\mathcal{S}}\frac{1}{1 + e^{-\gamma w_t - \bar{\mathbf{w}}_t^\top \bar{\mathbf{x}}_i}}\cdot e^{-2\bar{\mathbf{w}}_t^\top \bar{\mathbf{x}}_i}\cdot\|\bar{\mathbf{x}}_i\|_2 \qquad \text{by triangle inequality}$$

$$\leq e^{-2\gamma w_t}\cdot\sum_{i\in\mathcal{S}}\frac{1}{1 + e^{-\gamma w_t - \bar{\mathbf{w}}_t^\top \bar{\mathbf{x}}_i}}\cdot e^{-2\bar{\mathbf{w}}_t^\top \bar{\mathbf{x}}_i} \qquad \text{since } \|\bar{\mathbf{x}}_i\|_2 \leq 1 \text{ by Assumption 2}$$

$$\leq e^{-2\gamma w_t}\cdot\sum_{i\in\mathcal{S}}e^{-2\bar{\mathbf{w}}_t^\top \bar{\mathbf{x}}_i}$$

$$\leq e^{-2\gamma w_t}\cdot\left(\sum_{i\in\mathcal{S}}e^{-\bar{\mathbf{w}}_t^\top \bar{\mathbf{x}}_i}\right)^2$$

$$= e^{-2\gamma w_t}\cdot G(\bar{\mathbf{w}}_t)^2$$

$$\leq e^{-2\gamma w_t}\cdot G_{\max}^2. \qquad \text{by Definition 3}$$

The ($\heartsuit$) term can be bounded by

$$(\heartsuit) = \left\|\sum_{i\notin\mathcal{S}}\frac{e^{-w_t x_i - \bar{\mathbf{w}}_t^\top \bar{\mathbf{x}}_i}}{1 + e^{-w_t x_i - \bar{\mathbf{w}}_t^\top \bar{\mathbf{x}}_i}}\cdot\bar{\mathbf{x}}_i\right\|_2$$

$$\leq \sum_{i\notin\mathcal{S}}\frac{e^{-w_t x_i - \bar{\mathbf{w}}_t^\top \bar{\mathbf{x}}_i}}{1 + e^{-w_t x_i - \bar{\mathbf{w}}_t^\top \bar{\mathbf{x}}_i}}\cdot\|\bar{\mathbf{x}}_i\|_2 \qquad \text{by triangle inequality}$$

$$\leq \sum_{i\notin\mathcal{S}}\frac{e^{-w_t x_i - \bar{\mathbf{w}}_t^\top \bar{\mathbf{x}}_i}}{1 + e^{-w_t x_i - \bar{\mathbf{w}}_t^\top \bar{\mathbf{x}}_i}} \qquad \text{since } \|\bar{\mathbf{x}}_i\|_2 \leq 1 \text{ by Assumption 2}$$

$$\leq \sum_{i\notin\mathcal{S}}e^{-w_t x_i - \bar{\mathbf{w}}_t^\top \bar{\mathbf{x}}_i}$$

$$\leq e^{-\theta w_t}\cdot\sum_{i\notin\mathcal{S}}e^{-\bar{\mathbf{w}}_t^\top \bar{\mathbf{x}}_i} \qquad x_i \geq \theta > \gamma \text{ for } i\notin\mathcal{S}$$

$$= e^{-\theta w_t}\cdot H(\bar{\mathbf{w}})$$

$$\leq e^{-\theta w_t} \cdot H_{\max}. \qquad\qquad\qquad \text{by Definition 3}$$

Bringing the bounds on the ($\clubsuit$) and ($\heartsuit$) into (25), we obtain

$$\left\| \nabla_{\bar{\mathbf{w}}} L(w_t, \bar{\mathbf{w}}_t) - e^{-\gamma w_t} \cdot \nabla G(\bar{\mathbf{w}}_t) \right\|_2 \leq e^{-2\gamma w_t} \cdot G_{\max}^2 + e^{-\theta w_t} \cdot H_{\max}, \quad t \geq 0.$$

We have shown the first conclusion. The second conclusion follows from the first conclusion: for every $\mathbf{v} \in \mathbb{R}^{d-1}$,

$$\begin{aligned}
\langle \bar{\mathbf{v}}, \nabla_{\bar{\mathbf{w}}} L(w_t, \bar{\mathbf{w}}_t) \rangle &= e^{-\gamma w_t} \cdot \langle \bar{\mathbf{v}}, \nabla G(\bar{\mathbf{w}}_t) \rangle + \langle \bar{\mathbf{v}}, \nabla_{\bar{\mathbf{w}}} L(w_t, \bar{\mathbf{w}}_t) - e^{-\gamma w_t} \cdot \nabla G(\bar{\mathbf{w}}_t) \rangle \\
&\leq e^{-\gamma w_t} \cdot \langle \bar{\mathbf{v}}, \nabla G(\bar{\mathbf{w}}_t) \rangle + \|\bar{\mathbf{v}}\|_2 \cdot \|\nabla_{\bar{\mathbf{w}}} L(w_t, \bar{\mathbf{w}}_t) - e^{-\gamma w_t} \cdot \nabla G(\bar{\mathbf{w}}_t)\|_2 \\
&\leq e^{-\gamma w_t} \cdot \langle \bar{\mathbf{v}}, \nabla G(\bar{\mathbf{w}}_t) \rangle + \|\bar{\mathbf{v}}\|_2 \cdot \left( e^{-2\gamma w_t} \cdot G_{\max}^2 + e^{-\theta w_t} \cdot H_{\max} \right).
\end{aligned}$$

We have completed the proof. $\qquad\qquad\qquad\qquad\qquad\qquad\qquad\qquad\qquad\qquad\qquad\qquad \square$

**Lemma B.9** (A gradient norm bound). *Suppose Assumptions 1, 2, and 3 hold. Then it holds that*

$$\left\| \nabla_{\bar{\mathbf{w}}} L(w_t, \bar{\mathbf{w}}_t) \right\|_2 \leq e^{-\gamma w_t} \cdot (G_{\max} + H_{\max}), \quad t \geq 0.$$

*Proof.* The inequality is because:

$$\begin{aligned}
\left\| \nabla_{\bar{\mathbf{w}}_t} L(w_t, \bar{\mathbf{w}}_t) \right\|_2 &= \left\| \sum_{i=1}^{n} \frac{e^{-w_t x_i - \bar{\mathbf{w}}_t^\top \bar{\mathbf{x}}_i}}{1 + e^{-w_t x_i - \bar{\mathbf{w}}_t^\top \bar{\mathbf{x}}_i}} \cdot \bar{\mathbf{x}}_i \right\|_2 \\
&\leq \sum_{i=1}^{n} \frac{e^{-w_t x_i - \bar{\mathbf{w}}_t^\top \bar{\mathbf{x}}_i}}{1 + e^{-w_t x_i - \bar{\mathbf{w}}_t^\top \bar{\mathbf{x}}_i}} \cdot \|\bar{\mathbf{x}}_i\|_2 && \text{by triangle inequality} \\
&\leq \sum_{i=1}^{n} \frac{e^{-w_t x_i - \bar{\mathbf{w}}_t^\top \bar{\mathbf{x}}_i}}{1 + e^{-w_t x_i - \bar{\mathbf{w}}_t^\top \bar{\mathbf{x}}_i}} && \text{since } \|\bar{\mathbf{x}}_i\|_2 \leq 1 \text{ by Assumption 2} \\
&\leq \sum_{i=1}^{n} e^{-w_t x_i - \bar{\mathbf{w}}_t^\top \bar{\mathbf{x}}_i} \\
&\leq e^{-\gamma w_t} \cdot \sum_{i=1}^{n} e^{-\bar{\mathbf{w}}_t^\top \bar{\mathbf{x}}_i} && \text{since } x_i \geq \gamma \text{ for } i \in [n] \\
&= e^{-\gamma w_t} \cdot \left( G(\bar{\mathbf{w}}_t) + H(\bar{\mathbf{w}}_t) \right) \\
&\leq e^{-\gamma w_t} \cdot (G_{\max} + H_{\max}).
\end{aligned}$$

$\qquad\qquad\qquad\qquad\qquad\qquad\qquad\qquad\qquad\qquad\qquad\qquad\qquad\qquad\qquad\qquad \square$

The next lemma shows that the function value is "non-increasing" ignoring higher order terms.

**Lemma B.10** (A modified descent lemma). *Suppose Assumptions 1, 2, and 3 hold. Then it holds that*

$$G(\bar{\mathbf{w}}_{t+1}) \leq G(\bar{\mathbf{w}}_t) + 2(\eta + \eta^2) \cdot G_{\max} \cdot \left( G_{\max}^2 + H_{\max}^2 \right) \cdot \left( e^{-2\gamma w_t} + e^{-\theta w_t} \right), \quad t \geq 0.$$

*As a direct consequence of the above and Lemma B.6, it holds that*

$$G(\bar{\mathbf{w}}_{t+k}) \leq G(\bar{\mathbf{w}}_t) + c_0 \cdot 2(1 + \eta) G_{\max} \cdot \left( (t-1)^{-1} + (t-1)^{1-\frac{\theta}{\gamma}} \right), \quad k \geq 0, \ t \geq 1,$$

*where $\theta/\gamma > 1$ by Definition 1 and $c_0$ is a polynomial on $\left\{ e^\eta, e^n, e^{1/b}, \frac{1}{\eta}, \frac{1}{\theta - \gamma}, \frac{1}{\gamma}, e^{\theta/\gamma} \right\}$ and is independent of $t$, given by*

$$c_0 := \eta \cdot \left( G_{\max}^2 + H_{\max}^2 \right) \cdot \frac{\theta}{\theta - \gamma} \cdot \max \left\{ \left( \frac{2}{\eta \gamma^2} \right)^2, \left( \frac{2}{\eta \gamma^2} \right)^{\theta/\gamma} \right\}.$$

*Proof.* Note that

$$\|\nabla^2 G(\bar{\mathbf{w}})\|_2 = \left\| \sum_{i \in \mathcal{S}} e^{-\bar{\mathbf{w}}^\top \bar{\mathbf{x}}_i} \mathbf{x}_i \mathbf{x}_i^\top \right\|_2$$

$$\leq \sum_{i \in \mathcal{S}} e^{-\bar{\mathbf{w}}^\top \bar{\mathbf{x}}_i} \|\mathbf{x}_i\|_2^2 \qquad\qquad \text{by triangle inequality}$$

$$\leq \sum_{i \in \mathcal{S}} e^{-\bar{\mathbf{w}}^\top \bar{\mathbf{x}}_i} \qquad\qquad \text{since } \|\bar{\mathbf{x}}_i\|_2 \leq 1 \text{ by Assumption 2}$$

$$= G(\bar{\mathbf{w}}).$$

Recall that $\|\bar{\mathbf{w}}_t\|_2 \leq W_{\max}$. So we have

$$\sup_t \|\nabla^2 G(\bar{\mathbf{w}}_t)\|_2 \leq \sup_{\|\bar{\mathbf{w}}\|_2 \leq W_{\max}} \|\nabla^2 G(\bar{\mathbf{w}})\|_2 \leq \sup_{\|\bar{\mathbf{w}}\|_2 \leq W_{\max}} G(\bar{\mathbf{w}}) =: G_{\max}. \qquad (26)$$

Then we can apply Taylor's theorem to obtain that

$$G(\bar{\mathbf{w}}_{t+1}) \leq G(\bar{\mathbf{w}}_t) + \langle \nabla G(\bar{\mathbf{w}}_t), \ \bar{\mathbf{w}}_{t+1} - \bar{\mathbf{w}}_t \rangle + \frac{G_{\max}}{2} \cdot \|\bar{\mathbf{w}}_{t+1} - \bar{\mathbf{w}}_t\|_2^2 \qquad\qquad \text{by (26)}$$

$$= G(\bar{\mathbf{w}}_t) - \eta \cdot \langle \nabla G(\bar{\mathbf{w}}_t), \ \nabla_{\bar{\mathbf{w}}} L(w_t, \bar{\mathbf{w}}_t) \rangle + \frac{G_{\max}}{2} \cdot \|\nabla_{\bar{\mathbf{w}}} L(w_t, \bar{\mathbf{w}}_t)\|_2^2.$$

Next we use Lemma B.8 with $\mathbf{v} = -\nabla G(\bar{\mathbf{w}}_t)$ to get The cross-term is bounded by

$$- \langle \nabla G(\bar{\mathbf{w}}_t), \ \nabla_{\bar{\mathbf{w}}} L(w_t, \bar{\mathbf{w}}_t) \rangle$$

$$\leq -e^{-\gamma w_t} \cdot \left\|\nabla G(\bar{\mathbf{w}}_t)\right\|_2^2 + \|\nabla G(\bar{\mathbf{w}}_t)\|_2 \cdot \left(e^{-2\gamma w_t} \cdot G_{\max}^2 + e^{-\theta w_t} \cdot H_{\max}\right)$$

$$\leq -e^{-\gamma w_t} \cdot \left\|\nabla G(\bar{\mathbf{w}}_t)\right\|_2^2 + G_{\max} \cdot \left(e^{-2\gamma w_t} \cdot G_{\max}^2 + e^{-\theta w_t} \cdot H_{\max}\right). \qquad\qquad \text{by (26)}$$

Using the above and the gradient norm bound from Lemma B.9, we get that

$$G(\bar{\mathbf{w}}_{t+1}) \leq G(\bar{\mathbf{w}}_t) - \eta e^{-\gamma w_t} \cdot \left\|\nabla G(\bar{\mathbf{w}}_t)\right\|_2^2$$

$$+ \eta e^{-2\gamma w_t} \cdot G_{\max}^3 + \eta e^{-\theta w_t} \cdot G_{\max} \cdot H_{\max} + \eta^2 e^{-2\gamma w_t} \cdot (G_{\max} + H_{\max})^2$$

$$\leq G(\bar{\mathbf{w}}_t) + \eta e^{-2\gamma w_t} \cdot G_{\max}^3 + \eta e^{-\theta w_t} \cdot G_{\max} \cdot H_{\max} + \eta^2 e^{-2\gamma w_t} \cdot (G_{\max} + H_{\max})^2$$

$$\leq G(\bar{\mathbf{w}}_t) + 2(\eta + \eta^2) \cdot G_{\max} \cdot \left(G_{\max}^2 + H_{\max}^2\right) \cdot \left(e^{-2\gamma w_t} + e^{-\theta w_t}\right),$$

where in the last inequality we use that $G_{\max} \geq G_{\min} \geq 1$ by Definition 3.

From the above we have

$$G(\bar{\mathbf{w}}_{t+k}) \leq G(\bar{\mathbf{w}}_t) + 2(\eta + \eta^2) \cdot G_{\max} \cdot \left(G_{\max}^2 + H_{\max}^2\right) \cdot \sum_{s=t}^{s+k} \left(e^{-2\gamma w_s} + e^{-\theta w_s}\right). \qquad (27)$$

The summation is small by Lemma B.6, because

$$\sum_{s=t}^{s+k} \left(e^{-2\gamma w_s} + e^{-\theta w_s}\right)$$

$$\leq \sum_{s=t}^{s+k} \left(\frac{2}{2 + \eta\gamma^2 \cdot s}\right)^2 + \sum_{s=t}^{s+k} \left(\frac{2}{2 + \eta\gamma^2 \cdot s}\right)^{\frac{\theta}{\gamma}} \qquad\qquad \text{by Lemma B.6}$$

$$\leq \left(\frac{2}{\eta\gamma^2}\right)^2 \cdot \sum_{s=t}^{s+k} s^{-2} + \left(\frac{2}{\eta\gamma^2}\right)^{\frac{\theta}{\gamma}} \cdot \sum_{s=t}^{s+k} s^{-\frac{\theta}{\gamma}}$$

$$\leq \left(\frac{2}{\eta\gamma^2}\right)^2 \cdot (t-1)^{-1} + \left(\frac{2}{\eta\gamma^2}\right)^{\frac{\theta}{\gamma}} \cdot \frac{(t-1)^{1-\frac{\theta}{\gamma}}}{\frac{\theta}{\gamma} - 1} \qquad\qquad \text{by integral inequality}$$

$$\leq \max\left\{\left(\frac{2}{\eta\gamma^2}\right)^2, \left(\frac{2}{\eta\gamma^2}\right)^{\theta/\gamma}\right\} \cdot \frac{\theta}{\theta - \gamma} \cdot \left((t-1)^{-1} + (t-1)^{1-\frac{\theta}{\gamma}}\right).$$

Inserting the above into (27) completes the proof. $\qquad\qquad\qquad\qquad\qquad\qquad\qquad\qquad\square$

We now prove the convergence of the iterates projected on the non-separable subspace.

**Lemma B.11** (Convergence on the non-separable subspace). *Suppose Assumptions 1, 2, and 3 hold. Then it holds that*

$$G(\bar{\mathbf{w}}_T) - G(\bar{\mathbf{w}}_*) \leq \frac{c_1}{\log(T)}, \quad T \geq 3,$$

*where $c_1 > 0$ is a polynomial on $\left\{ e^\eta, e^n, e^{1/b}, \frac{1}{\eta}, \frac{1}{\theta - \gamma}, \frac{1}{\gamma}, e^{\theta/\gamma} \right\}$ and is independent of $T$.*

*Proof.* The proof is conducted in several steps.

**Step 1: one-step function value bound.** Observe that

$$\|\bar{\mathbf{w}}_{t+1} - \bar{\mathbf{w}}_*\|_2^2 = \|\bar{\mathbf{w}}_t - \bar{\mathbf{w}}_*\|_2^2 + 2 \cdot \langle \bar{\mathbf{w}}_t - \bar{\mathbf{w}}_*, \bar{\mathbf{w}}_{t+1} - \bar{\mathbf{w}}_t \rangle + \|\bar{\mathbf{w}}_{t+1} - \bar{\mathbf{w}}_t\|_2^2$$
$$= \|\bar{\mathbf{w}}_t - \bar{\mathbf{w}}_*\|_2^2 - 2\eta \cdot \langle \bar{\mathbf{w}}_t - \bar{\mathbf{w}}_*, \nabla_{\bar{\mathbf{w}}} L(w_t, \bar{\mathbf{w}}_t) \rangle + \eta^2 \cdot \|\nabla_{\bar{\mathbf{w}}} L(w_t, \bar{\mathbf{w}}_t)\|_2^2.$$

For the cross-term, we apply Lemma B.8 with $\mathbf{v} = -(\bar{\mathbf{w}}_t - \bar{\mathbf{w}}_*)$ to obtain

$$- \langle \bar{\mathbf{w}}_t - \bar{\mathbf{w}}_*, \nabla_{\bar{\mathbf{w}}} L(w_t, \bar{\mathbf{w}}_t) \rangle$$
$$\leq -e^{-\gamma w_t} \cdot \langle \bar{\mathbf{w}}_t - \bar{\mathbf{w}}_*, \nabla G(\bar{\mathbf{w}}_t) \rangle + \|\bar{\mathbf{w}}_t - \bar{\mathbf{w}}_*\|_2 \cdot \left( e^{-2\gamma w_t} \cdot G_{\max}^2 + e^{-\theta w_t} \cdot H_{\max} \right)$$
$$\leq -e^{-\gamma w_t} \cdot \langle \bar{\mathbf{w}}_t - \bar{\mathbf{w}}_*, \nabla G(\bar{\mathbf{w}}_t) \rangle + (W_{\max} + \|\mathbf{w}_*\|_2) \cdot \left( e^{-2\gamma w_t} \cdot G_{\max}^2 + e^{-\theta w_t} \cdot H_{\max} \right)$$
$$\leq -e^{-\gamma w_t} \cdot \langle \bar{\mathbf{w}}_t - \bar{\mathbf{w}}_*, \nabla G(\bar{\mathbf{w}}_t) \rangle + 2W_{\max} \cdot \left( e^{-2\gamma w_t} \cdot G_{\max}^2 + e^{-\theta w_t} \cdot H_{\max} \right),$$

where the second inequality is by Lemma B.3, and the last inequality is by Lemma B.4. Using the above and the gradient norm bound from Lemma B.9, we get that

$$\|\bar{\mathbf{w}}_{t+1} - \bar{\mathbf{w}}_*\|_2^2 \leq \|\bar{\mathbf{w}}_t - \bar{\mathbf{w}}_*\|_2^2 - 2\eta e^{-\gamma w_t} \cdot \langle \bar{\mathbf{w}}_t - \bar{\mathbf{w}}_*, \nabla G(\bar{\mathbf{w}}_t) \rangle$$
$$+ 4\eta \cdot W_{\max} \cdot \left( e^{-2\gamma w_t} \cdot G_{\max}^2 + e^{-\theta w_t} \cdot H_{\max} \right) \tag{28}$$
$$+ \eta^2 \cdot e^{-2\gamma w_t} \cdot (G_{\max} + H_{\max})^2.$$

By the convexity of $G(\cdot)$, we have

$$\langle \bar{\mathbf{w}}_t - \bar{\mathbf{w}}_*, \nabla G(\bar{\mathbf{w}}_t) \rangle \geq G(\bar{\mathbf{w}}_t) - G(\bar{\mathbf{w}}_*). \tag{29}$$

So we get

$$2\eta e^{-\gamma w_t} \cdot \left( G(\bar{\mathbf{w}}_t) - G(\bar{\mathbf{w}}_*) \right) \leq 2\eta e^{-\gamma w_t} \cdot \langle \bar{\mathbf{w}}_t - \bar{\mathbf{w}}_*, \nabla G(\bar{\mathbf{w}}_t) \rangle \quad \text{by (29)}$$
$$\leq \|\bar{\mathbf{w}}_t - \bar{\mathbf{w}}_*\|_2^2 - \|\bar{\mathbf{w}}_{t+1} - \bar{\mathbf{w}}_*\|_2^2$$
$$+ 4\eta \cdot W_{\max} \cdot \left( e^{-2\gamma w_t} \cdot G_{\max}^2 + e^{-\theta w_t} \cdot H_{\max} \right)$$
$$+ \eta^2 \cdot e^{-2\gamma w_t} \cdot (G_{\max} + H_{\max})^2 \quad \text{by (28)}$$
$$\leq \|\bar{\mathbf{w}}_t - \bar{\mathbf{w}}_*\|_2^2 - \|\bar{\mathbf{w}}_{t+1} - \bar{\mathbf{w}}_*\|_2^2$$
$$+ 6\eta \cdot W_{\max} \cdot \left( G_{\max}^2 + H_{\max}^2 \right) \cdot \left( e^{-2\gamma w_t} + e^{-\theta w_t} \right), \tag{30}$$

where we use $\eta \leq W_{\max} := \max\{4n/b, \eta n^2/b\} + \eta n$ in the last inequality.

**Step 2: the sum of function values stays bounded.** Observe that

$$\sum_{t=2}^{T} \left( e^{-2\gamma w_t} + e^{-\theta w_t} \right) \leq \sum_{t=2}^{T} \left( \frac{2}{2 + \eta\gamma^2 \cdot t} \right)^2 + \sum_{t=2}^{T} \left( \frac{2}{2 + \eta\gamma^2 \cdot t} \right)^{\frac{\theta}{\gamma}} \quad \text{by Lemma B.6}$$
$$\leq \left( \frac{2}{\eta\gamma^2} \right)^2 \cdot \sum_{t=2}^{T} t^{-2} + \left( \frac{2}{\eta\gamma^2} \right)^{\frac{\theta}{\gamma}} \cdot \sum_{t=2}^{T} t^{-\frac{\theta}{\gamma}}$$
$$\leq \left( \frac{2}{\eta\gamma^2} \right)^2 \cdot 1 + \left( \frac{2}{\eta\gamma^2} \right)^{\frac{\theta}{\gamma}} \cdot \frac{1}{\theta/\gamma - 1}$$
$$\leq \max \left\{ \left( \frac{2}{\eta\gamma^2} \right)^2, \left( \frac{2}{\eta\gamma^2} \right)^{\theta/\gamma} \right\} \cdot \frac{\theta}{\theta - \gamma}. \tag{31}$$

Taking telescope summation over (30), we obtain

$$\sum_{t=2}^{T} 2\eta e^{-\gamma w_t} \cdot \left( G(\bar{\mathbf{w}}_t) - G(\bar{\mathbf{w}}_*) \right)$$

$$\leq \|\bar{\mathbf{w}}_2 - \bar{\mathbf{w}}_*\|_2^2 - \|\bar{\mathbf{w}}_{T+1} - \bar{\mathbf{w}}_*\|_2^2$$

$$+ 6\eta \cdot W_{\max} \cdot (G_{\max}^2 + H_{\max}^2) \cdot \sum_{t=2}^{T} \left( e^{-2\gamma w_t} + e^{-\theta w_t} \right) \qquad \text{by (30)}$$

$$\leq 2W_{\max} + 6\eta \cdot W_{\max} \cdot (G_{\max}^2 + H_{\max}^2) \cdot \max\left\{ \left( \frac{2}{\eta\gamma^2} \right)^2, \left( \frac{2}{\eta\gamma^2} \right)^{\theta/\gamma} \right\} \cdot \frac{\theta}{\theta - \gamma} \qquad \text{by (31)}$$

$$= 2W_{\max} + 18W_{\max} \cdot c_0,$$

where

$$c_0 := \eta \cdot (G_{\max}^2 + H_{\max}^2) \cdot \frac{\theta}{\theta - \gamma} \cdot \max\left\{ \left( \frac{2}{\eta\gamma^2} \right)^2, \left( \frac{2}{\eta\gamma^2} \right)^{\theta/\gamma} \right\}$$

is a constant (a polynomial on $\left\{ e^\eta, e^n, e^{1/b}, \frac{1}{\eta}, \frac{1}{\theta - \gamma}, \frac{1}{\gamma}, e^{\theta/\gamma} \right\}$ and is independent of $t$) defined in Lemma B.10.

**Step 3: function value decreases, approximately.**   For $T \geq t \geq 1$, we have

$$G(\bar{\mathbf{w}}_T) \leq G(\bar{\mathbf{w}}_t) + c_0 \cdot 2(1 + \eta) G_{\max} \cdot \left( (t-1)^{-1} + (t-1)^{1 - \frac{\theta}{\gamma}} \right), \qquad \text{by Lemma B.10}$$

which implies that

$$2\eta e^{-\gamma w_t} \cdot \left( G(\bar{\mathbf{w}}_T) - G(\bar{\mathbf{w}}_*) \right)$$

$$\leq 2\eta e^{-\gamma w_t} \cdot \left( G(\bar{\mathbf{w}}_t) - G(\bar{\mathbf{w}}_*) \right) + 2\eta e^{-\gamma w_t} \cdot c_0 \cdot 2(1 + \eta) G_{\max} \cdot \left( (t-1)^{-1} + (t-1)^{1 - \frac{\theta}{\gamma}} \right)$$

$$\leq 2\eta e^{-\gamma w_t} \cdot \left( G(\bar{\mathbf{w}}_t) - G(\bar{\mathbf{w}}_*) \right)$$

$$+ 2\eta \cdot \frac{2}{2 + \eta\gamma^2 \cdot t} \cdot c_0 \cdot 2(1 + \eta) G_{\max} \cdot \left( (t-1)^{-1} + (t-1)^{1 - \frac{\theta}{\gamma}} \right) \qquad \text{by Lemma B.6}$$

$$\leq 2\eta e^{-\gamma w_t} \cdot \left( G(\bar{\mathbf{w}}_t) - G(\bar{\mathbf{w}}_*) \right) + \frac{8(1 + \eta)c_0}{\gamma^2} \cdot \left( (t-1)^{-2} + (t-1)^{-\frac{\theta}{\gamma}} \right). \qquad (32)$$

**Step 4: the last function value is small.**   Taking summation of (32) over $t = 2, \ldots T$, we get

$$\sum_{t=2}^{T} 2\eta e^{-\gamma w_t} \cdot \left( G(\bar{\mathbf{w}}_T) - G(\bar{\mathbf{w}}_*) \right)$$

$$\leq \sum_{t=2}^{T} 2\eta e^{-\gamma w_t} \cdot \left( G(\bar{\mathbf{w}}_t) - G(\bar{\mathbf{w}}_*) \right) + \frac{8(1 + \eta)c_0}{\gamma^2} \cdot \sum_{t=2}^{T} \left( (t-1)^{-2} + (t-1)^{-\frac{\theta}{\gamma}} \right)$$

$$\leq \sum_{t=2}^{T} 2\eta e^{-\gamma w_t} \cdot \left( G(\bar{\mathbf{w}}_t) - G(\bar{\mathbf{w}}_*) \right) + \frac{8(1 + \eta)c_0}{\gamma^2} \cdot \left( 2 + 1 + \frac{1}{\theta/\gamma - 1} \right)$$

$$\leq 2W_{\max} + 18W_{\max} \cdot c_0 + \frac{8(1 + \eta)c_0}{\gamma^2} \cdot \frac{3\theta}{\theta - \gamma}. \qquad \text{by (31)}$$

We also have

$$\sum_{t=2}^{T} e^{-\gamma w_t} \geq \frac{1}{e\eta\gamma^2 G_{\max} + e\eta\gamma H_{\max}} \cdot \sum_{t=2}^{T} \frac{1}{t+1} \qquad \text{by Lemma B.7}$$

$$\geq \frac{1}{e\eta\gamma^2 G_{\max} + e\eta\gamma H_{\max}} \cdot \left( \log(T + 1) - \log(3) \right)$$

Putting these together, we get

$$G(\bar{\mathbf{w}}_T) - G(\bar{\mathbf{w}}_*) \leq \left(2W_{\max} + 18W_{\max} \cdot c_0 + \frac{8(1+\eta)c_0}{\gamma^2} \cdot \frac{3\theta}{\theta - \gamma}\right) \cdot \frac{e\eta\gamma^2 G_{\max} + e\eta\gamma H_{\max}}{\log(T+1) - \log(3)},$$

where

$$c_0 := \eta \cdot (G_{\max}^2 + H_{\max}^2) \cdot \frac{\theta}{\theta - \gamma} \cdot \max\left\{\left(\frac{2}{\eta\gamma^2}\right)^2, \left(\frac{2}{\eta\gamma^2}\right)^{\theta/\gamma}\right\}$$

is a polynomial on $\left\{e^\eta, e^n, e^{1/b}, \frac{1}{\eta}, \frac{1}{\theta-\gamma}, \frac{1}{\gamma}, e^{\theta/\gamma}\right\}$. So for $T \geq 3$, we have

$$G(\bar{\mathbf{w}}_T) - G(\bar{\mathbf{w}}_*) \leq \frac{1}{\log(T)} \cdot c_1,$$

where $c_1$ is a polynomial on $\left\{e^\eta, e^n, e^{1/b}, \frac{1}{\eta}, \frac{1}{\theta-\gamma}, \frac{1}{\gamma}, e^{\theta/\gamma}\right\}$ and is independent of $T$. □

## C   Proofs Missing from the Main Paper

### C.1   Proof of Theorem 4.1

*Proof of Theorem 4.1.* Theorem 4.1 is a consequence of our analysis in Appendix B.

(C) is because of Lemma B.3.

(B) is because of Lemma B.6.

(D) is because of Lemma B.11.

(A) is because of the following:

$$\begin{aligned}
L(\mathbf{w}_t) &= \sum_{i=1}^n \log(1 + \exp(-w_t x_i - \mathbf{w}_t^\top \mathbf{x}_i)) \\
&\leq \sum_{i=1}^n \exp(-w_t x_i - \mathbf{w}_t^\top \mathbf{x}_i) \\
&\leq \exp(-w_t \cdot \gamma) \cdot \sum_{i=1}^n \exp(-\mathbf{w}_t^\top \mathbf{x}_i) \\
&\leq c/\log(t),
\end{aligned}$$

where the last inequality is because that

$$\exp(-w_t \cdot \gamma) \leq \frac{2}{2 + \eta\gamma^2 \cdot t}$$

by Lemma B.6 and that $\sum_{i=1}^n \exp(-\mathbf{w}_t^\top \mathbf{x}_i)$ is uniformly bounded by a constant by Definition 3. □

### C.2   Proof of Theorem 4.2

*Proof of Theorem 4.2.* The GD iterates can be written as

$$w_{t+1} = w_t + \eta\gamma \cdot e^{-\gamma w_t} \cdot \left(e^{-\bar{w}_t} + e^{\bar{w}_t}\right), \tag{33}$$

$$\bar{w}_{t+1} = \bar{w}_t - \eta e^{-\gamma w_t} \cdot \left(e^{-\bar{w}_t} - e^{\bar{w}_t}\right). \tag{34}$$

We claim that: for every $t \geq 0$,

1. $w_t \geq 0$.

2. $|\bar{w}_t| \geq 1$.

3. $|\bar{w}_t| \geq 2\gamma w_t$.

We prove the claim by induction. For $t = 0$, it holds by assumption. Now suppose that the claim holds for $t$ and consider the case of $t + 1$.

1. $w_{t+1} \geq 0$ holds since $w_{t+1} \geq w_t$ by (33) and $w_t \geq 0$ by the induction hypothesis.

2. $|\bar{w}_{t+1}| \geq 1$ holds because

$$
\begin{aligned}
|\bar{w}_{t+1}| &\geq \eta e^{-\gamma w_t} \cdot |e^{-\bar{w}_t} - e^{\bar{w}_t}| - |\bar{w}_t| \quad &&\text{by (34)} \\
&\geq \eta e^{-\gamma w_t} \cdot \frac{e^{|\bar{w}_t|}}{2} - |\bar{w}_t| \quad &&\text{since } |\bar{w}_t| \geq 1 \text{ and that } e^t - e^{-t} \geq \frac{e^t}{2} \text{ for } t \geq 1 \\
&\geq 2e^{|\bar{w}_t| - \gamma w_t} - |\bar{w}_t| \quad &&\text{since } \eta \geq 4 \\
&\geq 2e^{|\bar{w}_t|/2} - |\bar{w}_t| \quad &&\text{since } \frac{|\bar{w}_t|}{2} \geq \gamma w_t \\
&\geq 1. \quad &&\text{since } 2e^{t/2} \geq t + 1 \text{ for } t \in \mathbb{R}
\end{aligned}
\tag{35}
$$

3. To prove that $|\bar{w}_{t+1}| \geq 2\gamma w_t$, first observe that

$$
\begin{aligned}
w_{t+1} &= w_t + \eta\gamma \cdot e^{-\gamma w} \cdot \left(e^{-\bar{w}} + e^{\bar{w}}\right) \\
&\leq w_t + \eta\gamma \cdot e^{-\gamma w} \cdot 2 \cdot e^{|\bar{w}_t|}.
\end{aligned}
\tag{36}
$$

Then we have

$$
\begin{aligned}
&|\bar{w}_{t+1}| - 2\gamma w_{t+1} \\
&\geq \eta e^{-\gamma w_t} \cdot \frac{e^{|\bar{w}_t|}}{2} - |\bar{w}_t| - 2\gamma\left(w_t + \eta\gamma \cdot e^{-\gamma w} \cdot 2 \cdot e^{|\bar{w}_t|}\right) \quad &&\text{by (35) and (36)} \\
&= \frac{\eta}{2} \cdot (1 - 8\gamma^2) \cdot e^{|\bar{w}_t| - \gamma w_t} - |\bar{w}_t| - 2\gamma w_t \\
&\geq e^{|\bar{w}_t| - \gamma w_t} - |\bar{w}_t| - 2\gamma w_t \quad &&\text{since } \eta \geq 4 \geq 2/(1 - 8\gamma^2) \\
&\geq e^{|\bar{w}_t| - \gamma w_t} - |\bar{w}_t| \quad &&\text{since } w_t \geq 0 \\
&\geq e^{|\bar{w}_t|/2} - |\bar{w}_t| \quad &&\text{since } \frac{|\bar{w}_t|}{2} \geq \gamma w_t \\
&\geq 0. \quad &&\text{since } e^{t/2} \geq t \text{ for } t \in \mathbb{R}
\end{aligned}
$$

We have completed the induction.

Finally, we prove the claims in Theorem 4.2 using the above results.

(B) is because of

$$
w_{t+1} \geq w_t + \eta\gamma \cdot e^{-\gamma w_t}
$$

from (33).

We have already proved (C) by induction.

To show (D), without lose of generality, let us assume $\bar{w}_t \geq 0$, then

$$
\begin{aligned}
\bar{w}_{t+1} &\leq \bar{w}_t - \eta e^{-\gamma w_t} \cdot |e^{-\bar{w}_t} - e^{\bar{w}_t}| \quad &&\text{by (34)} \\
&\leq \bar{w}_t - \eta e^{-\gamma w_t} \cdot \frac{e^{|\bar{w}_t|}}{2} \quad &&\text{since } |\bar{w}_t| \geq 1 \text{ and that } e^t - e^{-t} \geq \frac{e^t}{2} \text{ for } t \geq 1 \\
&\leq \bar{w}_t - 2e^{|\bar{w}_t| - \gamma w_t} \quad &&\text{since } \eta \geq 4 \\
&\leq \bar{w}_t - 2e^{|\bar{w}_t|/2} \quad &&\text{since } \frac{|\bar{w}_t|}{2} \geq \gamma w_t \\
&\leq -1. \quad &&\text{since } 2e^{t/2} \geq t + 1 \text{ for } t \in \mathbb{R}
\end{aligned}
$$

We can repeat the above argument to show that $\bar{w}_{t+1} > 0$ if $\bar{w}_t \leq 0$.

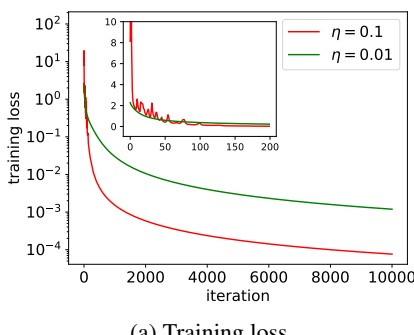
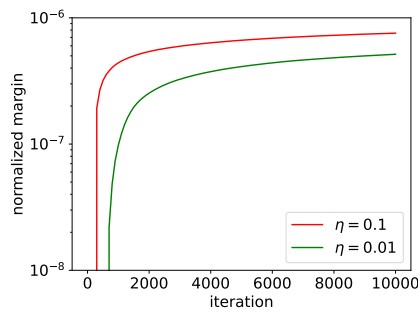

(a) Training loss

(b) Normalized margin for homogenous network

Figure 3: The behaviors of GD for optimizing a homogenous neural network. The experiment setting follows that of Figure 1 in the main paper, except that we disable all bias parameters so that the neural network is homogenous [Lyu and Li, 2020]. The sub-figures (a) and (b) report the training loss and the margin along the GD trajectories, respectively. Here, we follow Lyu and Li [2020] and measure the normalized margin by $\min_{(\mathbf{x},y)} \left( f(\mathbf{x}; \mathbf{w}_t)[y] - \max_{y' \neq y} f(\mathbf{x}; \mathbf{w}_t)[y'] \right) / \|\mathbf{w}_t\|_2^L$, where $f(\cdot; \cdot)$ refers to the homogenous neural network, $L = 4$ is the depth of the homogenous neural network, $\mathbf{w}_t$ is the weight at the $t$-th GD step, and $\min_{(\mathbf{x},y)}$ is taken with respect to all training data.

To show (A), we apply $w_t \to \infty$ and that $|\bar{w}_t| \geq 2\gamma w_t$:

$$L(w_t, \bar{w}_t) = e^{-\gamma w_t} \cdot \left( e^{-\bar{w}_t} + e^{\bar{w}_t} \right)$$
$$\geq e^{-\gamma w_t} \cdot e^{|\bar{w}_t|}$$
$$\geq e^{\gamma w_t} \to \infty.$$

We have completed all the proofs. $\qquad\square$

## D   Experimental Setups

**Neural network experiments.**  We randomly sample $1,000$ data from the MNIST[3] dataset as the training set and use the remaining data as the test set. The feature vectors are normalized such that each feature is within $[-1, 1]$.

We use a fully connected network with the following structure

$$784 \to \mathtt{ReLU} \to 500 \to \mathtt{ReLU} \to 500 \to \mathtt{ReLU} \to 500 \to \mathtt{ReLU} \to 10.$$

The network is initialized with Kaiming initialization. We use the cross-entropy loss.

We consider constant-stepsize GD with two types of stepsizes, $\eta = 0.1$ and $\eta = 0.01$.

The results are presented in Figure 1.

**Logistic regression experiments.**  We randomly sample $1,000$ data with labels "0" and "8" from the MNIST dataset as the training set. The feature vectors are normalized such that each feature is within $[-1, 1]$.

We use a linear model without bias. So the number of parameters is $784$. The model is initialized from zero. We use the binary cross-entropy loss, i.e., the logistic loss.

We consider constant-stepsize GD with various stepsizes.

The results are presented in Figure 2.

**Additional experiments.**  We conduct additional experiments on the margin maximization effect of large stepsize GD on a homogenous neural network. Results are presented in Figure 3.

---

[3] http://yann.lecun.com/exdb/mnist/

