# OpenReview forum: "Implicit Bias of Gradient Descent for Logistic Regression at the Edge of Stability"
_NeurIPS.cc/2023/Conference — NeurIPS 2023 spotlight_

### Official Review · Reviewer_QkkA · 2023-06-27

**Soundness:** 4 excellent
**Presentation:** 4 excellent
**Contribution:** 4 excellent
**Rating:** 8
**Confidence:** 3

**Summary:**

The authors study gradient descent for logistic regression on linearly separable data, with a focus on large stepsizes, and in particular stepsizes in the edge of stability (EoS) regime.
With the logistic loss, they show that for *any* positive stepsize, the empirical loss is minimized as the number of GD steps increases, and the limit direction maximises the margin. Explicit rates are given.
The authors also provide counterexamples for the exponential loss: if the stepsize is too large, the loss diverges and the iterates cannot converge to the maxmargin direction.



**Strengths:**

The authors study a hot topic (Edge of Stability) on an overall very well studied problem (logistic regression).
In these lines of work, the results are rather strong: proving convergence for any stepsize, beyond those that provide a nice descent lemma, seemed like a challenging problem, and this forms a strong contribution.

The other contributions (comparison between logistic and exp. losses, , the technical analysis) are also independently of interest.

Overall, the paper is neatly written and easy to follow.

**Weaknesses:**

There are no main concerns or weaknesses to this paper, in terms of results or presentation.
However, if I had to give one, I would say that since the implicit bias is the same whatever the stepsize is (it is thus the same for infinitesimal  ones and for stepsizes at the EoS), Theorem 1 extends previous results on the implicit bias of logistic regression, but does not give insight onto what happens at the edge of stability, and what drives generalization to increase. Could this be commented by the authors ?

Also, here are some recent references that study the EoS phenomenon on other problems, and that should be mentionned in the related works (they are all quite recent):

In contrast to the logistic regression problem studied in this submission, large stepsizes on regression problem lead to different solutions for GD and SGD when at the edge of stability:
*(S) GD over Diagonal Linear Networks: Implicit Regularisation, Large Stepsizes and Edge of Stability*, Even et al. 2022.

The following paper also studies *scalar* linear networks (a scalar parametered by a product $xy$), as done in Ahn et al. 2022a,b and Zhu et al. 2022 cited in the submission, but in a more precise way: *Gradient Descent Monotonically Decreases the Sharpness of Gradient Flow Solutions in Scalar Networks and Beyond*.

Finally, *SGD with Large Step Sizes Learns Sparse Features*, Andriushchenko et al. 2022 study large stepsizes SGD and generalization (more experimental piece of work).

**Questions:**

I have no further questions.

**Limitations:**

No such limitations.

---

> ### Author Rebuttal · Authors · 2023-08-06
>
> We appreciate your strong support! Thank you for pointing out additional related papers. We will make sure to cite and discuss them properly in the revision. We address your question below.
>
> ---
> **Q1**. “...since the implicit bias is the same whatever the stepsize is (it is thus the same for infinitesimal ones and for stepsizes at the EoS), Theorem 1 extends previous results on the implicit bias of logistic regression, but does not give insight onto what happens at the edge of stability, and what drives generalization to increase. Could this be commented by the authors?”
>
> **A1**. Good question. Indeed, our paper only characterizes the behavior of large stepsize GD after an (exponentially) long time, where its implicit bias is the same as that of small stepsize GD. To understand the (generalization) benefits of large stepsize, we need to further consider the polynomial-time behaviors of large/small stepsize GD and what exactly happens at the edge of stability. We conjecture that there is a different implicit bias induced by large and small stepsize GD when the optimization length is only polynomially large. We also think the new analysis techniques presented in this work will be useful for understanding the polynomial-time implicit bias of constant-stepsize GD. We believe this is an important future direction and will comment on this in the revision.

---

> > ### Comment · Reviewer_QkkA · 2023-08-17
> >
> > I thank the authors for their answer. I keep my very good opinion of this paper unchanged.

---

### Official Review · Reviewer_PfDy · 2023-07-04

**Soundness:** 4 excellent
**Presentation:** 4 excellent
**Contribution:** 3 good
**Rating:** 7
**Confidence:** 3

**Summary:**

This paper analyzes the convergence and implicit bias of logistic regression on linearly separable datasets with arbitrarily large step sizes.  The paper proves that gradient descent converges _for any_ constant step size, even those which result in non-decreasing loss curves.  Similar to the regime of small step sizes, gradient descent converges along the direction of the max-margin linear separator.

In contrast to logistic loss, the paper proves that for exponential loss there exist datasets and step sizes where the optimizer fails to converge, shedding light on why logistic loss may be preferable in practice over exponential loss.

**Strengths:**

- It's interesting (and far from obvious a priori) that GD with _any_ constant step size will converge.
- The separation between logistic and exponential losses sheds light on why the former is preferred in practice (since both losses are equivalent under the small step size regime studied in previous works)
- The separable / non-separable orthogonal decomposition seems like it might be useful for other analyses of logistic regression on separable datasets.

**Weaknesses:**

- Minor: would Assumption 3 hold if the dataset were the last-layer features from a neural net has undergone neural collapse, or would such a dataset not satisfy assumption 3?

**Questions:**

- Could you include some numerical examples of convergence at extremely large step sizes?  I am curious to see what the loss curve looks like.

- The following recent paper should be discussed:

Chunrui Liu, Wei Huang, and Richard Yi Da Xu.  "Implicit Bias of Deep Learning in the Large Learning Rate Phase: A Data Separability Perspective."  Applied Sciences, 2023.  https://www.mdpi.com/2076-3417/13/6/3961

The paper does not seem to be on arxiv, and the venue is not a standard one ... nevertheless, the paper is quite related.

In their Proposition 1, that paper argues that for logistic regression on _non-separable datasets_ (or, more generally, for any strongly convex optimization problem), gradient descent fails to converge for large enough step sizes.

---

> ### Author Rebuttal · Authors · 2023-08-06
>
> We appreciate your affirmative comments! Thank you for pointing out the paper by [Liu, Huang, and Xu, 2023]. We will properly cite and discuss the paper in the revision. We will address your other questions as follows.
>
> ---
> **Q1**. “Minor: would Assumption 3 hold if the dataset were the last-layer features from a neural net has undergone neural collapse, or would such a dataset not satisfy assumption 3?”
>
> **A1**. Under neural collapse, the (last-layer) features tend to be close to their class mean. Even so, Assumption 3 holds under weak conditions, e.g., when features can be viewed as the class-mean feature plus noise from a continuous distribution. This is because Assumption 3 holds with probability 1 as long as the data/feature is generated from a continuous distribution [Soudry et al. 2018].
>
> ---
> **Q2**. “Could you include some numerical examples of convergence at extremely large step sizes? I am curious to see what the loss curve looks like.”
>
> **A2**. Please see Figure 3 in the attached rebuttal PDF for the suggested experiment. We tested two extremely large stepsizes, $\eta=10^2$ and $\eta=10^3$ in the logistic regression setting. Figure 3 is consistent with our results.

---

### Official Review · Reviewer_iDYE · 2023-07-10

**Soundness:** 3 good
**Presentation:** 3 good
**Contribution:** 3 good
**Rating:** 7
**Confidence:** 4

**Summary:**

The paper studies the convergence of gradient descent with large step sizes beyond the classical stepsize suggest by smoothness-based convergence. Across the iterates, the risk here exhibits non-monotonic behaviour and yet converges in the long range.  To show these phenomenon, the paper rigorously studies the gradient descent with large step-size in the case of logistic regression with linearly separable data.

**Strengths:**

a) The paper studies logistic regression in the case of separable data. Even in this simpler case, the authors prove a very strong result. It is shown that gradient descent with **any** constant stepsize converges.

b) For gradient descent with any step-size, the implicit bias towards the max $\ell_2$-margin solution is shown.

c) A novel proof technique was introduced to handle any stepsize and it was clearly presented with a simple example. In contrast to existing works, the implicit bias is shown which leads to convergence in risk.

**Weaknesses:**

a) The proof is tailor made for linear regression and does not work beyond that for any convex functions.

**Questions:**

a) In the case of exponential loss, it is shown that it can diverge with a large stepsize for a simple example. Do you think there exists a sub-class of problems where exponential loss can converge with any step size?



**Limitations:**

---

> ### Author Rebuttal · Authors · 2023-08-06
>
> Thank you for supporting our paper! We address your comments in the following.
>
> ---
> **Q1**. “The proof is tailor made for linear regression and does not work beyond that for any convex functions.”
>
> **A1**. We respectfully point out that our analysis is for logistic regression rather than linear regression. Our analysis can be extended to other (convex) generalized linear models where the loss functions behave similarly to the logistic loss. For example, our proofs hold (with minor revisions on constant factors) when the logistic loss
> $$ \ell(t) = \log(1+e^{-t}) $$
> is replaced by a variant of the *Exponential Linear Unit*:
> $$ \tilde{\ell}(t) = \begin{cases} a\cdot (e^{-t} -1), & t \ge 0; \\\ -t, & t<0, \end{cases}$$
> where $a>0$ is a constant.
> The key properties of the logistic loss in our analysis are 1) the gradient is bounded and 2) the function value of a negative input dominates that of a positive input. We will provide a detailed explanation of these alongside our proof in the revision.
>
> ---
> **Q2**. “In the case of exponential loss, it is shown that it can diverge with a large stepsize for a simple example. Do you think there exists a sub-class of problems where exponential loss can converge with any step size?”
>
> **A2**. In special situations, GD can converge under exponential loss with any constant stepsize. For example, it holds if the dataset is such that the feature vectors are orthogonal to each other. One can prove this claim by noticing that the parameter updates can be perfectly decoupled along each feature direction.

---

> > ### Comment · Reviewer_iDYE · 2023-08-17
> > **Reply to Author Rebuttal**
> >
> > Q1. I apologize for the overlook it should have been logistic instead of linear.
> >
> > Thanks for the clarifications.

---

### Official Review · Reviewer_pcDT · 2023-07-11

**Soundness:** 3 good
**Presentation:** 3 good
**Contribution:** 2 fair
**Rating:** 5
**Confidence:** 3

**Summary:**

This research investigates the convergence and implicit bias of constant-stepsize GD for logistic regression on linearly separable data in the EoS regime. The authors assert that despite local oscillations, the logistic loss can be minimized by GD with constant stepsize over a long time scale. 	Experiments on neural networks and logistic regression verify the theory and illustrate the non-monotonic convergence and implicit bias of GD.

**Strengths:**

- Provides novel theoretical analysis of GD allowing constant step sizes, complementing existing theories requiring small step sizes. This better matches practical deep learning where large steps are used.
- Establishes implicit bias results for GD with any constant step size. Shows GD aims to maximize the margin regardless of step size.
- The analysis technique of decomposing GD iterates into orthogonal components is innovative. It allows handling large step sizes that cause loss oscillations.
- Experiments on neural networks and logistic regression align with theory and illustrate non-monotonic convergence and implicit bias.
- Shows logistic loss is better behaved than exponential loss for large-stepsize GD, providing practical insights.

**Weaknesses:**

- The analysis is limited to logistic regression on linearly separable data. Extending it to more complex models like neural networks is not clear.
- Some assumptions made, like linearly separable data and non-degenerate support vectors, may be restrictive and not hold in practice. For example, wrongly labeled samples are common.

**Questions:**

How to define linearly separable in neural network literature?

It is clear that when you use a large learning rate, the neural network goes far away from the linear regime. How can you make a compensation to use the current model to understand the non-linearity in neural networks?


**Limitations:**

This work makes valuable theoretical contributions but has limitations in terms of connection to practice. Extending the analysis to address these limitations remains an open challenge.

---

> ### Author Rebuttal · Authors · 2023-08-06
>
> Thank you for appreciating our contributions including a novel theory, new techniques, and practical insights. We address your concerns below.
>
> ---
> **Q1**. “The analysis is limited to logistic regression on linearly separable data. Extending it to more complex models like neural networks is not clear.”
>
> **A1**. Extending our results to more complicated models is an important future work direction. However, even in the simple logistic regression setting, there is no such kind of result before our work for GD with large stepsizes. Without a full understanding of large stepsize GD for the simplest possible problem — linear model under logistic loss — it seems unlikely that one can achieve this in more complicated models such as neural networks. As our work is the first work on the implicit bias of large stepsize GD, we believe we have taken an important step in this direction and the contribution of our work is very significant.
>
> ---
> **Q2**. “Some assumptions made, like linearly separable data and non-degenerate support vectors, may be restrictive and not hold in practice. For example, wrongly labeled samples are common.”
>
> **A2**. We emphasize that Assumptions 1 and 3 are standard conditions made by prior papers, e.g., the pioneering work on implicit bias by Soudry et al. [2018]. Therefore, our novel theory development is not based on more restrictive assumptions compared to prior works. Extending our results to cover wrongly labeled samples is an interesting direction, which we will comment on in the revision as future work!
>
> ---
> **Q3**. “How to define linearly separable in neural network literature?”
>
> **A3**. The linear separability condition can be generalized to a “separability” condition in (homogenous) neural network cases, that is, there exists a (homogenous) neural network that achieves $100\\%$ training accuracy (see, e.g., Section 1.1 in [Lyu and Li, 2019]).
>
> ---
> **Q4**. “It is clear that when you use a large learning rate, the neural network goes far away from the linear regime. How can you make a compensation to use the current model to understand the non-linearity in neural networks?”
>
> **A4**. For homogenous neural networks, which are far away from a linear model, Lyu and Li [2019] and Nacson et al. [2019a] prove that small stepsize GD converges to a max-margin solution (under a revised definition of “margin”). We conjecture that our results for large stepsize GD for logistic regression can also be extended to homogenous neural networks in a similar way. In Figure 4 in the attached rebuttal PDF, we provide numerical results on a homogenous neural network, which are consistent with Lyu and Li [2019] and support our conjecture. We believe this is an important future direction and will discuss this in detail in the revision.

---

> > ### Comment · Reviewer_pcDT · 2023-08-20
> >
> > Thank you for your comments. I will maintain my rating.

---

> > > ### Author Response · Authors · 2023-08-20
> > > **Thank you for your response**
> > >
> > > We hope all of your concerns have been addressed. Please do let us know if there is anything else that requires our clarification.

---

### Official Review · Reviewer_rZLX · 2023-07-23

**Soundness:** 3 good
**Presentation:** 3 good
**Contribution:** 3 good
**Rating:** 6
**Confidence:** 4

**Summary:**

This work investigates the implicit bias in linear regression with logistic loss with large learning rates. It shows the convergence result of logistic loss for any learning rates, as long as an counter-example of exponential loss to show the gap between these two losses.

**Strengths:**

1. The motivation and setting are good: large learning rates (i.e., Edge of Stability) are a rising topic recently. As pointed out by the authors, it requires reconsideration of previous results of implicit regularization with small learning rates. To my knowledge, this work is the first to redo one of the bias results in the standard setting.

2. The gap between logistic and exponential losses is good: typically implicit-bias works handle these two similarly, but this work shows that logistic loss can contain arbitrarily large learning rates while exponential loss has an upper bound.

**Weaknesses:**

1. The covered setting is somehow simple in lack of detailed discussions of what impact logistic loss + large learning rates have on more complicated settings possibly. For instance, does the max-margin bias in deep homogenous model (Lyu and Li) hold any more? This kind of impact could be verified with visualized experiments, with the theoretical development left as future works.

**Questions:**

As above in weakness.

Typos & minor suggestions:
1. below 299: it is $\nabla G$ instead of $G$ in the inner product.

2. in Figure 2, the bound of $2/\eta=0.2$ can be plotted in a more clear way, such as log.

---

> ### Author Rebuttal · Authors · 2023-08-06
>
> Thank you for your comments, suggestions, and positive evaluation. Please find our answers to your questions below.
>
> ---
>
> **Q1**. “The covered setting is somehow simple in lack of detailed discussions of what impact logistic loss + large learning rates have on more complicated settings possibly. For instance, does the max-margin bias in deep homogenous model (Lyu and Li) hold any more? This kind of impact could be verified with visualized experiments, with the theoretical development left as future works.”
>
> **A1**. Please see Figure 4 in the attached rebuttal PDF for the suggested experiments on a homogenous deep neural network. Figure 4 is consistent with the results of [Lyu and Li, 2019], showing a margin-maximization tendency of GD with both large and small stepsizes for homogenous models. We will incorporate these into the paper in more detail.
>
> We agree that extending our results from linear model to more complicated models such as homogenous models is an important future work direction. However, even in the simple logistic regression setting, there is no such kind of result before our work. Without a full understanding of large stepsize GD for the simplest possible problem — linear model under logistic loss — it seems unlikely that one can achieve this in more complicated models. As our work is the first work on the implicit bias of large stepsize GD, we believe we have taken an important step in this direction and the contribution of our work is very significant.
>
> ---
>
> **Q2**. “Typos & minor suggestions…”
>
> **A2**. We will fix the typo in the equation below line 299. We have changed the linear scale of the y-axis in Figure 2(b) to a logarithmic scale to improve readability. See Figure 3(b) in the attached rebuttal PDF. We are happy to incorporate any additional suggestions you have to improve our paper!

---

### Author Rebuttal · Authors · 2023-08-06

The attached rebuttal PDF contains the results of suggested experiments for logistic regression with extremely large stepsizes and homogenous neural networks.

---

### Decision · Program_Chairs · 2023-09-21

**Decision:**

Accept (spotlight)

**Comment:**

All reviewers deem the paper interesting. So I recommend accepting the paper as a spotlight.